# Optimal Statistical Rates for Decentralised Non-Parametric Regression with Linear Speed-Up

**Dominic Richards**
Department of Statistics
University of Oxford
24-29 St Giles', Oxford, OX1 3LB
dominic.richards@spc.ox.ac.uk

**Patrick Rebeschini**
Department of Statistics
University of Oxford
24-29 St Giles', Oxford, OX1 3LB
patrick.rebeschini@stats.ox.ac.uk

## Abstract

We analyse the learning performance of Distributed Gradient Descent in the context of multi-agent decentralised non-parametric regression with the square loss function when i.i.d. samples are assigned to agents. We show that if agents hold sufficiently many samples with respect to the network size, then Distributed Gradient Descent achieves optimal statistical rates with a number of iterations that scales, *up to a threshold*, with the inverse of the spectral gap of the gossip matrix divided by the number of samples owned by each agent raised to a problem-dependent power. The presence of the threshold comes from statistics. It encodes the existence of a "big data" regime where the number of required iterations does *not* depend on the network topology. In this regime, Distributed Gradient Descent achieves optimal statistical rates with the *same* order of iterations as gradient descent run with all the samples in the network. Provided the communication delay is sufficiently small, the distributed protocol yields a *linear* speed-up in runtime compared to the single-machine protocol. This is in contrast to decentralised optimisation algorithms that do not exploit statistics and only yield a linear speed-up in graphs where the spectral gap is bounded away from zero. Our results exploit the statistical concentration of quantities held by agents and shed new light on the interplay between statistics and communication in decentralised methods. Bounds are given in the standard non-parametric setting with source/capacity assumptions.

## 1 Introduction

In machine learning a canonical goal is to use training data sampled independently from an unknown distribution to fit a model that performs well on unseen data from the same distribution. With a loss function measuring the performance of a model on a data point, a common approach is to find a model that minimises the average loss on the training data with some form of *explicit* regularisation to control model complexity and avoid overfitting. Due to the increasingly large size of datasets and high model complexity, direct minimisation of the regularised problem is posing more and more computational challenges. This has led to growing interest in approaches that improve models incrementally using gradient descent methods [8], where model complexity is controlled through forms of *implicit/algorithmic* regularisation such as early stopping and step-size tuning [57, 58, 27].

The growth in the size of modern datasets has also meant that the coordination of multiple machines is often required to fit machine learning models. In the centralised server-clients setup, a single machine (server) is responsible to aggregate and disseminate information to other machines (clients) in what is an effective star topology. In some settings, such as ad-hoc wireless and peer-to-peer networks, network instability, bandwidth limitation and privacy concerns make centralised approaches less feasible. This has motivated research into scalable methods that can avoid the bottleneck

and vulnerability introduced by the presence of a central authority. Such solutions are called "decentralised", as no single entity is responsible for the collection and dissemination of information: machines communicate with neighbours in a network structure that encodes communication channels.

Since the early works [52, 53] to the more recent work [22, 34, 33, 23, 29, 30, 10, 18, 47, 31], problems in decentralised multi-agent optimisation have often been treated as a particular instance of consensus optimisation. In this framework, a network of machines or agents collaborate to minimise the average of functions held by individual agents, hence "reaching consensus" on the solution of the global problem. In this setting the performance of the chosen protocol naturally depends on the network topology, since to solve the problem each agent *has to* communicate and receive information from all other agents. In particular, the number of iterations required by decentralised iterative gradient methods typically scales with the inverse of the spectral gap of the communication matrix (a.k.a. gossip or consensus matrix) [18, 42, 43], which reflects the performance of gossip protocols in the problem of distributed averaging [9, 17, 44, 4].

Many distributed machine learning problems, in particular those involving empirical risk minimisation, have been framed in the context of consensus optimisation. However, as highlighted in [46] and more recently in [38], often these problems have more structure than consensus optimisation due to the statistical regularity of the data. When the agents' functions are the empirical risk of their local data, in the setting where the local data comes from the *same* unknown distribution (homogeneous setting), the functions held by each agent are similar to one another by the phenomenon of statistical concentration. In particular, in the limit of an infinite amount of data per agent, the local functions are the same and agents do *not* need to communicate to solve the problem. This phenomenon highlights the existence of a natural trade-off between statistics and communication. While statistical similarities of local objective functions and the statistics/communication trade-off have been investigated and exploited in centralised server-clients setup, typically in the analysis and design of divide-and-conquer schemes [60, 28, 20, 32, 26, 1, 62, 46, 45, 61, 2], only recently there has been some investigation into the interplay between statistics and communication/network-topology in the decentralised setting. The authors in [6] investigate the interplay between the spectral norm of the data-generating distribution and the inverse spectral gap of the communication matrix for Distributed Stochastic Gradient Descent in the case of strongly convex losses. As most of the literature on decentralised machine learning, this work also focuses on minimising the training error and not the test/prediction error (numerical experiments are given for the test error). Some works have investigated the performance on the test loss in the single-pass/online stochastic setting where agents use each data point only once. The authors in [37, 51] investigate a distributed regularised online learning setting [55] and obtain guarantees for a "multi-step" Distributed Stochastic Mirror Descent algorithm where agents reach consensus on their stochastic gradients in-between computation steps. The works [25] and [3] consider the performance of Distributed Stochastic Gradient Descent algorithms in the non-convex smooth case. They investigate the average performance of the agents over the network in terms of convergence to a stationary point of the test loss [19] and show that a linear speed-up in computational time can be achieved provided the number of samples seen, equivalently the number of iterations performed, exceeds the network size times the inverse of the spectral gap, each raised to a certain power. The work [38] seems to be the first to have considered minimisation of the test error in the multi-pass/offline stochastic setting that more naturally relates to the classical literature on consensus optimisation. The authors investigate stability of Distributed Stochastic Gradient Descent on the test error and show that for smooth and convex losses the number of iterations required to achieve optimal statistical rates scales with the inverse of the spectral gap of the gossip matrix, a term that captures the noise of the gradients' estimates, and a term that controls the statistical proximity of the local empirical losses.

## 1.1 Contributions

In this work we investigate the implicit-regularisation learning performance of full-batch Distributed Gradient Descent [33] on the test error in the context of non-parametric regression with the square loss function. In the homogeneous setting where agents hold independent and identically distributed data points, we investigate the choice of step size and number of iterations that guarantee each agent to achieve optimal statistical rates with respect to all the samples in the network. We build a theoretical framework that allows to directly and explicitly exploit the statistical concentration of quantities (i.e. batched gradients) held by agents. On the one hand, exploiting concentration yields savings on computation, i.e. it allows to achieve faster convergence rates compared to methods that do not exploit

concentration in their parameter tuning. On the other hand, it yields savings on communication, as it allows to take advantage of the trade-off between statistical power and communication costs. Firstly, we show that if agents hold sufficiently many samples with respect to the network size, then Distributed Gradient Descent achieves optimal statistical rates up to poly-logarithmic factors with a number of iterations that scales with the inverse of the spectral gap of the communication matrix divided by the number of samples owned by each agent raised to a problem-dependent power, up to a statistics-induced threshold. Previous results for decentralised iterative gradient schemes in the context of consensus optimisation do not take advantage of the statistical nature of decentralised empirical risk minimisation problems. In the statistical setting that we consider, these methods would require a larger number of iterations that scales only with respect to the inverse of the spectral gap. Secondly, we show that if agents additionally hold sufficiently many samples with respect to the inverse of the spectral gap, then the *same* order of iterations allows Distributed Gradient Descent and Single-Machine Gradient Descent (i.e. gradient descent run on a single machine that holds all the samples in the network) to achieve optimal statistical rates up to poly-logarithmic factors. Provided the communication delay is sufficiently small, this yields a *linear* speed-up in runtime over Single-Machine Gradient Descent, with a "single-step" method that performs a single communication round per local gradient descent step. Single-step methods that do not exploit concentration can only achieve a linear speed-up in runtime in graphs with spectral gap bounded away from zero, i.e. expanders or the complete graph. Our results demonstrate how the increased statistical similarity between the local empirical risk functions can make up for a decreased connectivity in the graph topology, showing that a linear speed-up in runtime can be achieved in *any* graph topology by exploiting concentration. To the best of our knowledge, we seem to be the first to isolate this type of phenomena.

We prove our results under the standard "source" and "capacity" assumptions in non-parametric regression. These assumptions relate, respectively, to the projection of the optimal predictor on the hypothesis space and to the effective dimension of this space [59, 12]. A contribution of this work is to show that proper tuning yields, up to poly-logarithmic terms, optimal non-parametric rates in decentralised learning. As far as we aware, in the distributed setting such guarantees have been established only for centralised divide-and-conquer methods [60, 28, 20, 32, 26].

To prove our results we build upon previous work for Single-Machine Gradient Descent applied to non-parametric regression, in particular the line of works [57, 40, 27]. Exploiting that in our setting the iterates of Distributed Gradient Descent can be written in terms of products of linear operators depending on the data held by agents, we decompose the excess risk into bias and sample variance terms for Single-Machine Gradient Descent plus an additional quantity that captures the error incurred by using a decentralised protocol over the communication network. We analyse this network error term by further decomposing it into a term that behaves similarly to the consensus error previously considered in [18, 33], and a new higher-order term. We control both terms by using the structure of the gradient updates, which allows us to analyse the interplay between statistics, via concentration, and network topology, via mixing of random walks related to the gossip matrix.

The work is structured as follows. Section 2 presents the setting, assumptions, and algorithm that we consider. Section 3 states the main convergence result and discusses implications from the point of view of statistics, computation and communication. Section 4 presents the error decomposition into bias, variance, and network error, and it illustrates the implicit regularisation strategy that we adopt. Section 5 highlights some of the features of our contribution in the light of future research directions. The appendix in the supplementary material is structured as follows. Section A includes some remarks about our results. Section B illustrates the main scheme of the proofs, highlighting the interplay between statistics and network topology. Section C contains the full details of the proofs.

## 2 Setup

In this section we describe the learning problem, assumptions and algorithm that we consider.

### 2.1 Learning problem: decentralised non-parametric least-squares regression

We adopt the setting used in [40, 27], which involves regression in abstract Hilbert spaces. This setting is of relevance for applications related to the Reproducing Kernel Hilbert Space (RKHS). See the work in [57] and references therein.

Let $H$ be a separable Hilbert Space with inner product and induced norm denoted by $\langle\,\cdot\,,\,\cdot\,\rangle_H$ and $\|\cdot\|_H$, respectively. Let $X \subseteq H$ be the input space and $Y \subset \mathbb{R}$ be the output space. Let $\rho$ be an unknown probability measure on $Z = X \times Y$, $\rho_X(\,\cdot\,)$ be the marginal on $X$, and $\rho(\,\cdot\,|x)$ be the conditional distribution on $Y$ given $x \in X$. Assume that there exists a constant $\kappa \in [1, \infty)$ so that

$$\langle x, x'\rangle_H \leq \kappa^2, \quad \forall x, x' \in X. \tag{1}$$

Let the network of agents be modelled by a simple, connected, undirected, finite graph $G = (V, E)$, with $|V| = n$ nodes joined by edges $E \subseteq V \times V$. Edges represent communication constraints: agents $v, w \in V$ can only communicate if they share an edge $(v, w) \in E$. We consider the homogeneous setting where each agent $v \in V$ is given $m$ data points $\mathbf{z}_v := \{\mathbf{x}_v, \mathbf{y}_v\}$ sampled independently from $\rho$, where $\mathbf{x}_v = \{x_{i,v}\}_{i=1,\dots,m}$ and $\mathbf{y}_v = \{y_{i,v}\}_{i=1,\dots,m}$, and each pair $(x_{i,v}, y_{i,v})$ is sampled from $\rho$. The problem under study is the minimisation of the test/prediction risk with the square loss:

$$\inf_{\omega \in H} \mathcal{E}(\omega), \qquad \mathcal{E}(\omega) = \int_{X \times Y} (\langle \omega, x\rangle_H - y)^2 d\rho(x, y), \tag{2}$$

The quality of an approximate solution $\widehat{\omega} \in H$ is measured by the excess risk $\mathcal{E}(\widehat{\omega}) - \inf_{\omega \in H} \mathcal{E}(\omega)$.

**Notation** Given a matrix $A \in \mathbb{R}^{n \times n}$, let $A_{vw}$ denote the $(v, w)$-th element and $A_v = (A_{vw})_{w=1,\dots,n}$ denote the $v$-th row. Let $O(\,\cdot\,)$ denote orders of magnitudes up to constants in $n$ and $m$, and $\widetilde{O}(\,\cdot\,)$ denote orders of magnitudes up to both constants and poly-logarithmic terms in $n$ and $m$. Let $\lesssim, \gtrsim, \simeq$ denote inequalities and equalities modulo constants and poly-logarithmic terms in $n, m$. We use the notation $a \vee b = \max\{a, b\}$ and $a \wedge b = \min\{a, b\}$.

## 2.2 Assumptions

The assumptions that we consider are standard in non-parametric regression [27, 35]. The first assumption is a control on the even moments of the response.

**Assumption 1.** *There exist $M \in (0, \infty)$ and $\nu \in (1, \infty)$ such that for any $\ell \in \mathbb{N}$ we have $\int_Y y^{2\ell} d\rho(y|x) \leq \nu \ell! M^\ell \ \rho_X$-almost surely.*

Let $L^2(H, \rho_X)$ be the Hilbert space of square-integrable functions from $H$ to $\mathbb{R}$ with respect to $\rho_X$, with norm $\|f\|_\rho := (\int_X |f(x)|^2 d\rho_X(x))^{1/2}$. Let $\mathcal{L}_\rho : L^2(H, \rho_X) \to L^2(H, \rho_X)$ be the operator defined as $\mathcal{L}_\rho(f) := \int_X \langle x, \cdot\,\rangle_H f(x) d\rho_X(x)$. Under Assumption 1 the operator $\mathcal{L}_\rho$ can be proved to be in the class of positive trace operators [15], and therefore the $r$-th power $\mathcal{L}_\rho^r$, with $r \in \mathbb{R}$, can be defined by using spectral theory. Let us also define the operator $\mathcal{T}_\rho : H \to H$ as $\mathcal{T}_\rho := \int_X \langle x, \cdot\,\rangle_H x d\rho_X(x)$ and its operator norm $\|\mathcal{T}_\rho\| := \sup_{\omega \in H, \|\omega\|_H = 1} \|\mathcal{T}_\rho \omega\|_H$. The function minimising the expected squared loss (2) over all measurable functions $f : H \to \mathbb{R}$ is known to be the conditional expectation $f_\rho(x) := \int_Y y d\rho(y|x)$ for $x \in \mathcal{X}$. Let $H_\rho := \{f : X \to \mathbb{R} \,|\, \exists \omega \in H \text{ with } f(x) = \langle w, x\rangle_H, \rho_X\text{-almost surely}\}$ be the hypothesis space that we consider. The optimal $f_\rho$ may not be in $H_\rho$ as under Assumption 1 the space of functions searched $H_\rho$ is a subspace of $L^2(H, \rho_X)$. Let $f_H$ denote the projection of $f_\rho$ onto the closure of $H_\rho$ in $L^2(H, \rho_X)$. Searching for a solution to (2) is equivalent to searching for a linear function in $H_\rho$ that approximates $f_H$.

The following assumption quantifies how well the target function $f_H$ can be approximated in $H_\rho$.

**Assumption 2.** *There exist $r > 0$ and $R > 0$ such that $\|\mathcal{L}_\rho^{-r} f_H\|_\rho \leq R$.*

This assumption is often called the "source" condition [12]. Representing $f_H$ in the eigenspace of $\mathcal{L}_\rho$, this condition can be related to the rate at which the coefficients of this representation decay. The bigger $r$ is, the faster the decay, and more stringent the assumption is. In particular, if $r \geq 1/2$ then the target function is in the hypothesis space $f_H \in H_\rho$. The last assumption is on the capacity of the hypothesis space.

**Assumption 3.** *There exist $\gamma \in (0, 1], c_\gamma > 0$ such that $\mathrm{Tr}(\mathcal{L}_\rho(\mathcal{L}_\rho + \lambda I)^{-1}) \leq c_\gamma \lambda^{-\gamma}$ for all $\lambda > 0$.*

Assumption 3 relates to the effective dimension of the underlying regression problem [59, 12] and is often called the "capacity" assumption. This assumption is always satisfied for $\gamma = 1$ and $c_\gamma = \kappa^2$ since $\mathcal{L}_\rho$ is a trace class operator. This case is called the capacity-independent setting. Meanwhile, this assumption is satisfied for $\gamma \in (0, 1]$ if, for instance, the eigenvalues of $\mathcal{L}_\rho$, denoted by $\{\tau_i\}_{i \geq 1}$, decay sufficiently quickly, i.e. $\tau_i = O(i^{-1/\gamma})$. This case allows improved rates to be obtained. For more details on the interpretation of these assumptions we refer to the work in [40, 27, 35].

## 2.3 Algorithm: distributed gradient descent

We now describe the Distributed Gradient Descent algorithm [33] and its application to the problem of non-parametric regression. Let $P \in \mathbb{R}_{\geq 0}^{n \times n}$ be a symmetric doubly-stochastic matrix, i.e. $P = P^{\top}$ and $P\mathbf{1} = \mathbf{1}$ where $\mathbf{1} = (1, \ldots, 1) \in \mathbb{R}^n$ is the vector of all ones. Let $P$ be supported on the graph, i.e. for any $v \neq w$, $P_{vw} \neq 0$ only if $(v, w) \in E$. The matrix $P$ encodes local averaging on the network: when each agent has a real number represented by the vector $a = (a_v)_{v \in V} \in \mathbb{R}^n$, the vector $(Pa)_v = \sum_{w \in V} P_{vw} a_w$ for $v \in V$ encodes what each agent computes after taking a weighted average of its own and neighbours' numbers. Distributed Gradient Descent is implemented by communication on the network through the gossip matrix $P$. Initialised at $w_{1,v} = 0$ for $v \in V$, the iterates of the Distributed Gradient Descent are defined as follows, for $v \in V$ and $t \geq 1$:

$$\omega_{t+1,v} = \sum_{w \in V} P_{vw}\Big(\omega_{t,w} - \eta_t \frac{1}{m} \sum_{i=1}^{m} \big(\langle \omega_{t,w}, x_{i,w} \rangle_H - y_{i,w}\big) x_{i,w}\Big), \tag{3}$$

where $\{\eta_t\}_{t \geq 1}$ is the sequence of positive step sizes. The iterates (3) can be seen as a combination of two steps: first, each agent $w \in V$ performs a local gradient descent step $\omega_{t+1/2,w} = \omega_{t,w} - \eta_t \frac{1}{m} \sum_{i=1}^{m} \big(\langle \omega_{t,w}, x_{i,w} \rangle_H - y_{i,w}\big) x_{i,w}$; second, each agent performs local averaging through the consensus step[1] $\omega_{t+1,v} = \sum_{w \in V} P_{vw} \omega_{t+1/2,w}$. We treat gradient descent as a statistical device. We are interested in tuning the parameters of the algorithm to bound the expected value of the excess risk $\mathbf{E}[\mathcal{E}(\omega_{t+1,v})] - \inf_{\omega \in H} \mathcal{E}(\omega)$, where $\mathbf{E}[\,\cdot\,]$ denotes expectation with respect to the data $\{\mathbf{z}_v\}_{v \in V}$.

**Network dependence** Let $\sigma_2$ be the second largest eigenvalue in magnitude of the communication matrix $P$. Specifically, given the spectral decomposition of the gossip matrix $P = \sum_{l=1}^{n} \lambda_l u_l u_l^{\top}$ where $1 = \lambda_1 \geq \lambda_2 \geq, \ldots, \geq \lambda_n > -1$ are the ordered real eigenvalues of $P$ and $\{u_l\}_{l=1,\ldots,n}$ the associated eigenvectors, we have $\sigma_2 := \max\{|\lambda_2|, |\lambda_n|\}$. In many settings, the spectral gap scales with the size of the network raised to a certain power depending on the topology. For instance, supposing $G$ is a finite regular graph and the communication matrix is the random walk matrix, then the inverse of the spectral gap $(1 - \sigma_2)^{-1}$ scales as $\Theta(1)$ for a complete graph, $\Theta(n)$ for a grid, and $\Theta(n^2)$ for a cycle [14, 24, 18]. The question of designing gossip matrices $P$ that yield better (smaller) scaling for the quantity $(1 - \sigma_2)^{-1}$ has been investigated [56], and it has been found numerically that the rates mentioned above can not be improved unless lifted graphs are considered [44].

## 3 Main result: optimal statistical rates with linear speed-up in runtime

We now state and highlight the main contribution of this work in the context of decentralised statistical optimisation. The result that we are about to state in Theorem 1 showcases the interplay between statistics and communication that arise from the statistical regularities of the problem. This result shows the existence of a "big data" regime where Distributed Gradient Descent can achieve a linear (in the number of agents $n$) speed-up in runtime compared to Single-Machine Gradient Descent.

**Theorem 1.** *Let Assumptions 1, 2, 3 hold with $r \geq 1/2$ and $2r + \gamma > 2$. Let $t$ be the smallest integer greater than the quantity*

$$\underbrace{(nm)^{1/(2r+\gamma)}}_{\text{Single-Machine Iterations}} \times \begin{cases} \left(\dfrac{(nm)^{2r/(2r+\gamma)}}{m(1-\sigma_2)^{\gamma}}\right)^{1/\gamma} \vee 1 & \text{if } m \geq n^{2r/\gamma} \\ \dfrac{(nm)^{r/(2r+\gamma)}}{\sqrt{m}(1-\sigma_2)} & \text{otherwise} \end{cases}$$

*Let $\eta_s \equiv \eta = \dfrac{\kappa^{-2}(nm)^{1/(2r+\gamma)}}{t} \,\forall s \geq 1$. If $m \geq n^{\frac{2r+2+\gamma}{2r+\gamma-2}}$ and $n \geq 2(1+r)\log(\frac{n}{1-\sigma_2})$, then $\forall v \in V$:*

$$\mathbf{E}[\mathcal{E}(\omega_{t+1,v})] - \inf_{\omega \in H} \mathcal{E}(\omega) \leq C(nm)^{-2r/(2r+\gamma)},$$

*where $C$ depends on $\kappa^2, \|\mathcal{T}_{\rho}\|, M, \nu, r, R, \gamma, c_{\gamma}$, and polynomials of $\log(nm)$ and $\log(\frac{1}{1-\sigma_2})$.*

Theorem 1 shows that when agents are given sufficiently many samples ($m$) with respect to the number of agents ($n$), $m \geq n^{\frac{2r+2+\gamma}{2r+\gamma-2}}$, proper tuning of the step size and number of iterations (a form of implicit regularisation) allows Distributed Gradient Descent to recover the optimal statistical rate $O((nm)^{-2r/(2r+\gamma)})$ for $r \in (1/2, 1)$ [12] up to poly-logarithmic terms.

Single-Machine Gradient Descent run on all of the observations has been previously shown to reach optimal statistical accuracy with a number of iterations of the order $t_{\text{Single-Machine}} \sim O((nm)^{1/(2r+\gamma)})$ [27]. The number of iterations $t \equiv t_{\text{Distributed}}$ prescribed by Theorem 1 scales like $t_{\text{Single-Machine}}$ times a network-dependent factor that is a function of the inverse of the spectral gap $(1 - \sigma_2)^{-1}$. The fact that the number of iterations required to reach a prescribed level of error accuracy is inversely proportional to the spectral gap is a standard feature of iterative gradient methods applied to generic decentralised consensus optimisation problems [18, 42, 43]. This dependence encodes the fact that in the case of *generic* objective functions assigned to agents, agents *have to* share information with everyone to solve the global problem and minimise the sum of the local functions; hence, more iterations are required in graph topologies that are less well-connected. In the present homogeneous setting, however, the statistical nature of the problem allows to exploit concentration of random variables to characterise the existence of a (network-dependent) "big data" regime where the number of iterations does *not* depend on the network topology. The trade-off between statistics and communication is encoded by the dependence of the tuning parameters (stopping time and step size) on the number of samples $m$ assigned to each agent. Observe that the factor $\left(\frac{(nm)^{2r/(2r+\gamma)}}{m(1-\sigma_2)^{\gamma}}\right)^{1/\gamma} \vee 1$ is a decreasing function of $m$, up to the threshold 1. When $m \geq \frac{n^{2r/\gamma}}{(1-\sigma_2)^{2r+\gamma}} \vee n^{\frac{2r+2+\gamma}{2r+\gamma-2}}$ this factor becomes 1 and Theorem 1 guarantees that the *same* order of iterations allows both Distributed and Single-Machine Gradient Descent to achieve the optimal statistical rates up to poly-logarithmic factors. This regime represents the case when the increased statistical similarity between the local empirical risk functions assigned to each agent (increasing as a function of $m$, as described by the non-asymptotic Law of Large Numbers) makes up for the decreased connectivity in the graph topology (typically decreasing with the spectral gap $1 - \sigma_2$) to yield a linear speed-up in runtime over Single-Machine Gradient Descent when the communication delay between agents is sufficiently small. See Section 3.1 below.

The result of Theorem 1 depends on some other requirements which we now briefly discuss. The requirement $n \geq 2(1 + r) \log(\frac{n}{1-\sigma_2})$ is technical and arises from the need to perform sufficiently many iterations to reach the mixing time of the gossip matrix $P$, i.e. $t \gtrsim (1 - \sigma_2)^{-1}$. Noting that the number of iterations $t$ depends on the number of agents, samples and spectral gap. The requirement $2r + \gamma > 2$ relates to the difficulty of the estimation problem and is stronger than a similar condition seen for single-machine gradient methods where $2r + \gamma > 1$, see for instance the works [27, 35]. This requirement, alongside $m \geq n^{\frac{2r+2+\gamma}{2r+\gamma-2}}$, ensures that the higher-order error terms arising from considering a decentralised protocol decay sufficiently quickly with respect to the number of samples owned by agents $m$. The condition $m \geq n^{\frac{2r+2+\gamma}{2r+\gamma-2}}$ can be removed if the covariance operator $\mathcal{T}_\rho$ is assumed to be known to agents, which aligns with the additive noise oracle in single-pass Stochastic Gradient Descent [16] or fixed-design regression in finite-dimensional settings [21]. The condition $m \geq n^{2r/\gamma}$ corresponds to the case when the rate of concentration of the batched gradients held by agents (i.e. $1/m$) is faster than the optimal statistical rate, i.e. $\frac{1}{m} \leq (nm)^{-2r/(2r+\gamma)}$. This condition becomes more stringent (i.e. more data per agent is needed) as the problem becomes easier from a statistical point of view and $r$ and $1/\gamma$ increase (see discussion in Section 2.2). This is due to the fact that as $r$ and $1/\gamma$ increase, only the statistical rate improves while the rate of concentration in the network error stays the same, implying that more data is needed to balance the two terms.

## 3.1 Linear speed-up in runtime

Let gradient computations cost 1 unit of time and communication delay between agents be $\tau$ units of time.[2] Denote the number of iterations required by Single-Machine Gradient Descent and Distributed Gradient Descent to achieve the optimal statistical rate by $t_{\text{Single-Machine}}$ and $t_{\text{Distributed}}$, respectively. The speed-up in computational time obtained by running the distributed protocol over the single-machine protocol is of the order $\frac{t_{\text{Single-Machine}}}{t_{\text{Distributed}}} \frac{nm}{m+\tau+\text{Deg}(P)}$, where $\text{Deg}(P) = \max_{v \in V} |\{P_{vw} \neq 0, w \in V\}|$ is the maximum degree of the communication matrix $P$. Theorem 1 implies that when $m \geq$

$\frac{n^{2r/\gamma}}{(1-\sigma_2)^{2r+\gamma}} \vee n^{\frac{2r+2+\gamma}{2r+\gamma-2}}$ then $t_{\text{Distributed}} \sim t_{\text{Single-Machine}}$, and if $\tau + \text{Deg}(P)$ grows as $O(m)$ then the speed-up in computational time is of order $n$, linear in the number of agents. Classical "single-step" decentralised methods that alternate single communication rounds per local gradient computation, such as the methods inspired by [33], do not exploit concentration and have a runtime that scales with the inverse of the spectral gap, without any threshold. As a result, these methods only yield a linear speed-up in graphs with spectral gap bounded away from zero, i.e. expanders or the complete graph. See below for more details. On the other hand, "multi-step" methods that alternate multiple communication rounds per local gradient computation, such as the ones considered in [37, 51, 42, 43], display a runtime that scales with a factor of the form $m + \frac{\tau + \text{Deg}(P)}{1-\sigma_2}$ in our setting. Thus, while these methods can achieve a linear speed-up in any graph topology in the "big data" regime $m \gtrsim \frac{\tau + \text{Deg}(P)}{1-\sigma_2}$ without exploiting concentration, they require an additional amount of communication rounds that is network-dependent and scales with the inverse of the spectral gap. For a cycle graph, for instance, this means an extra $O(n^2)$ communication steps per iteration (or $O(n)$ for gossip-accelerated methods). Hence, classical decentralised optimisation methods that do not exploit concentration suffer from a trade-off between runtime and communication cost: if you reduce the first you increase the second, and viceversa. Our results show that single-step methods can achieve a linear speed-up in runtime in *any* graph topology by exploiting concentration: statistics allows to find a regime where it is possible to simultaneously have a linear speed-up in runtime without increasing communication.

**Comparison to single-step decentralised methods that do not exploit concentration**   Decentralised optimisation methods that do not consider statistical concentration rates in their parameter tuning can not exploit the statistics/communication trade-off encoded by the presence of the factor $(\frac{(nm)^{2r/(2r+\gamma)}}{m(1-\sigma_2)^{\gamma}})^{1/\gamma} \vee 1$ in Theorem 1, and they typically require a smaller step size and more iterations to achieve optimal statistical rates. The convergence rate typically achieved by classical consensus optimisation methods, e.g. [18], is recovered in Theorem 1 when $m = n^{2r/\gamma}$ as in this case the number of iterations required becomes $t \sim \frac{(nm)^{1/(2r+\gamma)}}{1-\sigma_2}$, which corresponds to $t_{\text{Single-Machine}}$ scaled by a certain power of $1/(1-\sigma_2)$ (in our setting the power is 1). This represents the setting where the choice of step size aligns with the choice in the single-machine case scaled by $(1-\sigma_2)$, and a linear speed-up occurs when $(1-\sigma_2)^{-1} = O(1)$. Since the network error is decreasing in $m$ in our case (due to concentration), larger step sizes can be chosen for $m > n^{2r/\gamma}$. Specifically, the single-machine step size is now scaled by $[(1-\sigma_2)(\frac{m}{n^{2r/\gamma}})^{1/(2r+\gamma)}] \vee 1$, yielding a linear speed-up when $(1-\sigma_2)^{-1} = O((\frac{m}{n^{2r/\gamma}})^{1/(2r+\gamma)})$, which, as $m$ increases, is a weaker requirement on the network topology over the standard consensus optimisation setting.

## 4  General result: error decomposition and implicit regularisation

Theorem 1 is a corollary of the next result, which explicitly highlights the interplay between statistics and network topology and the implicit regularisation role of the step size and number of iterations.

**Theorem 2.** *Let Assumptions 1, 2, 3 hold with $r \geq 1/2$. Let $\eta_s = \eta s^{-\theta}\ \forall s \geq 1$ with $\theta \in (0, 3/4)$ and $\eta \in (0, \kappa^{-2}]$. If $t/2 \geq \lceil \frac{(r+1)\log(t)}{1-\sigma_2}\rceil =: t^\star$, then for all $v \in V$, $\alpha \in [0, 1/2]$ and $\gamma' \in [1, \gamma]$:*

$$\mathbb{E}[\mathcal{E}(\omega_{t+1,v})] - \inf_{\omega \in H} \mathcal{E}(\omega)$$

$$\leq \left[ q_1(\eta t^{1-\theta})^{-2r} + q_2(nm)^{-2r/(2r+\gamma)}\left(1 \vee (nm)^{-2/(2r+\gamma)}(\eta t^{1-\theta})^2 \vee t^{-2}(\eta t^{1-\theta})^2\right)\right]\log^2(t) \quad (4)$$

$$+ q_3\frac{\log^2(n)\log^2(t^\star)}{m}\left(\eta^2 t^{-2r} \vee (m^{-1}(\eta t^\star)^{1+2\alpha}) \vee (\eta t^\star)^{\gamma'+2\alpha}\right) \quad (5)$$

$$+ q_4\frac{\log^4(n)\log^2(t)}{m^2}\left(1 \vee (\eta t^{1-\theta})^2 \vee t^{-2}(\eta t^{1-\theta})^4\right)\left((m^{-1}\eta t^{1-\theta}) \vee (\eta t^{1-\theta})^\gamma\right) \quad (6)$$

*where $q_1, q_2, q_3, q_4$ are all constants depending on $\kappa^2, \|\mathcal{T}_\rho\|, M, \nu, r, R, \gamma, c_\gamma$.*

The bound in Theorem 2 shows that the excess risk has been decomposed into three main terms, as detailed in Section B.1. The first term (4) corresponds to the error achieved by Single-Machine Gradient Descent run on all $nm$ samples. It consists of both bias and sample variance terms [27]. The second two terms (5) and (6) characterise the network error due to the use of a decentralised

protocol. These terms decrease with the number of samples $m$ owned by each agent. This captures the fact that, as agents are given samples from the *same* unknown distribution, agents are in fact solving the same learning problem and their local empirical loss functions concentrate to the same objective as $m$ increases. The decentralised error term is itself composed of two terms which decay at different rates with respect to $m$. The term in (5) is dominant and decays at the order of $\widetilde{O}(1/m)$. This can be interpreted as the consensus error seen in the works [33, 18] for instance. As in that setting, this quantity is also increasing with the step size $\eta$ and decreasing with the spectral gap of the communication matrix $1 - \sigma_2$, as encoded by $t^\star$. The term (6) decays at the faster rate of $\widetilde{O}(1/m^2)$. This is a higher-order error term that is not appearing in the error decomposition when the covariance operator $\mathcal{T}_\rho$ is assumed to be known to agents. This quantity arises from the interaction between the local averaging on the network through $P$ and what has been previously labelled as the "multiplicative" noise in the single-machine single-pass stochastic gradient setting for least squares [16], i.e. the empirical covariance operator interacting with the iterates at each step. Section B.2 provides a high-level illustration of the analysis of the Network Error terms (5) and (6).

The bound in Theorem 2 shows how the algorithmic parameters—step size and number of iterations—act as regularisation parameters for Distributed Gradient Descent, following what is seen in the single-machine setting. Theorem 1 demonstrates how optimal statistical rates can be recovered by tuning these parameters appropriately with respect to the network topology, network size, number of samples, and with respect to the estimation problem itself. The bound in Theorem 1 is obtained from the bound in Theorem 2 by first tuning the quantity $\eta t$ to the order $(nm)^{1/(2r+\gamma)}$ so that the bias and variance terms in (4) achieve the optimal statistical rate. This leaves the tuning of the remaining degree of freedom (say $\eta$) to ensure that also the network error achieves the optimal statistical rate. The high-level idea is the following. As $m$ increases, the network error is dominated by the term in (5) that is proportional to the factor $(\eta t^\star)^{\gamma' + 2\alpha}/m$. There are two ways to choose the largest possible step size $\eta$ to guarantee that this factor is $\widetilde{O}((nm)^{-2r/(2r+\gamma)})$, depending on whether the rate of concentration of the batched gradients held by agents is faster than the optimal statistical rate or not, i.e., whether $m \geq n^{2r/\gamma}$ is true or not (cf. Section 3). The two cases yield the factors $(\frac{(nm)^{2r/(2r+\gamma)}}{m(1-\sigma_2)^\gamma})^{1/\gamma} \vee 1$ and $\frac{(nm)^{r/(2r+\gamma)}}{\sqrt{m}(1-\sigma_2)}$ in Theorem 1, corresponding to the choice $\gamma' = \gamma$, $\alpha = 0$ and $\gamma' = 1$, $\alpha = 1/2$, respectively. If the concentration of the batched gradients held by agents fully compensates for the network error, i.e. $m \geq \frac{n^{2r/\gamma}}{(1-\sigma_2)^{2r+\gamma}}$, then $(\eta t^\star)^{\gamma' + 2\alpha}/m \simeq (nm)^{-2r/(2r+\gamma)}$ with a constant step size and $t_{\text{Distributed}} \sim t_{\text{Single-Machine}} \sim (nm)^{1/(2r+\gamma)}$, yielding the regime where a linear speed-up occurs. For more details on the parameters $\alpha, \gamma'$, see Lemma 8 in Appendix C.3.1.

# 5 Future directions

We highlight some of the features of our contribution and outline directions for future research.

**Non-parametric setting**  We prove bounds in the attainable case $r \geq 1/2$. The non-attainable case $r < 1/2$ is known to be more challenging [27], and it is natural to investigate to what extent our results can be extended to that setting. We consider the case $\gamma > 0$ which does not include the finite-dimensional setting $H = \mathbb{R}^d$, $\gamma = 0$, where the optimal rate is $O(d/(nm))$ [54]. While adapting our results to this setting requires minor modifications, optimal bounds would only hold for "easy" estimation problems with $r > 1$ due to the higher-order term in the network error. Improvements require getting better bounds on this term, potentially using a different learning rate.

**General loss functions**  The analysis that we develop is specific to the square loss, which yields the bias/variance error decomposition and allows to get explicit characterisations by expanding the squares. While the concentration phenomena that we exploit are generic, different techniques are required to extend our analysis to other losses, as in the single-machine setting. The statistical proximity of agents' functions in the finite-dimensional setting has been investigated in [38].

**Statistics/communication trade-off with sparse/randomised gossip**  In this work we show that when agents hold sufficiently many samples, then Distributed and Single-Machine Gradient Descent achieve the optimal statistical rate with the same order of iterations. This motivates balancing and trading off communication and statistics, e.g., investigating statistically robust procedures in settings when agents communicate with a subset of neighbours, either deterministically or randomly [9, 17, 4].

**Stochastic gradient descent and mini-batches** Our work exploits concentration of gradients around their means, so full-batch gradients (i.e. batches of size $m$) yield the concentration rate $1/m$. In single-machine learning, stochastic gradient descent [39] has been shown to achieve good statistical performance in a variety of settings while allowing for computational savings. Extending our findings to stochastic methods with appropriate mini-batch sizes is another venue for future investigation.

### Acknowledgments

Dominic Richards is supported by the EPSRC and MRC through the OxWaSP CDT programme (EP/L016710/1). Patrick Rebeschini is supported in part by the Alan Turing Institute under the EPSRC grant EP/N510129/1. We would like to thank Francis Bach, Lorenzo Rosasco and Alessandro Rudi for helpful discussions.

## Footnotes

[1] We note, while this assumes agents communicate infinite dimensional quantities in the general non-parametric setting, the framework we consider accommodates finite approximations of infinite dimensional quantities whilst accounting for the statistical precision [13].

[2] For details on this communication model as well as comparison to [50] see remarks within Appendix A.

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
