[Supplementary Material · neurips_2019_supplementary.pdf]

# A Remarks

In this section we present some remarks about our work.

**Alternative protocol**  The protocol investigated in [33] updates the iterates via $\omega_{t+1,v} = \sum_{w \in V} P_{vw}\omega_{t,w} - \eta_t \frac{1}{m} \sum_{i=1}^{m} (\langle \omega_{t,v}, x_{i,v} \rangle_H - y_{i,v}) x_{i,v}$. The original motivations for this protocol are that it is fully decentralised, that agents are only required communicate locally, and that it reduces to a distributed averaging consensus protocol when the gradient is zero. The protocol (3) that we consider preserves these properties while making the analysis easier. For a discussion on the difference between the two protocols we refer to [41].

**Network error**  The network error terms (5) and (6) track the error between the distributed protocol and the ideal single-machine protocol. In the case of a complete graph the deviation is zero so the network terms vanish and the convergence rates for Single-Machine Gradient Descent are recovered. Following the literature on decentralised optimisation, we present our final results (cf. Theorem 2) in terms of the spectral gap, so plugging in the spectral gap of a complete graph in the bound in Theorem 2 does not immediately yield the Single-Machine Gradient Descent result.

**Parameter tuning**  The choice of parameters in Theorem 1 depends on the quantities $r$ and $\gamma$ that are related to the estimation problem. In practice, these quantities are often unknown. In the single-machine setting, this lack of knowledge is typically addressed via cross-validation [48]. Investigating the design of decentralised cross-validation schemes is outside of the scope of this work and we leave it to future research. However, we highlight that as we consider implicit regularisation strategies and, in particular, early stopping, model complexity can be controlled with iteration time and this yields computational savings for cross-validation compared to methods that required to solve independent problem instances for different choices of parameters.

**Accelerated gossip**  Accelerated gossip schemes can also be considered to yield improved dependence on the network topology, depending on the amount of information agents have access to about the communication matrix $P$. Accelerated gossip can be achieved by replacing the matrix $P$ by a polynomial of appropriate order, e.g. $k$, leading to $\widetilde{P} := \sum_{\ell=1}^{k} \alpha_\ell P^\ell$. The weights $\{\alpha\}_{\ell=1,\dots,K}$ can be tuned to increase the spectral gap i.e. $(1 - \sigma_2(\widetilde{P}))^{-1} \leq (1 - \sigma_2)^{-1}$. We highlight that the algorithm that we consider only needs to have access to the number of nodes $n$ and the second largest eigenvalue in magnitude $\sigma_2$ of the matrix $P$. Within this framework, one can use Chebyshev polynomials to obtain the improved rate $(1 - \sigma_2(\widetilde{P}))^{-1/2}$, and more information on the spectrum of $P$ yields better rates on the transitive phase [11, 5].

**Additional requirements in Theorem 2**  Theorem 2 includes two additional requirements over single-machine gradient descent, which we briefly explain the origins of. The requirement $\theta \leq 3/4$ is purely cosmetic and serves to yield a cleaner bound. For more details, see the proof of Lemma 9 in Section C.3.2. The requirement $t/2 \geq \frac{(r+1)\log(t)}{1-\sigma_2}$, on the other hand, often arises when analysing Distributed Gradient Descent, see [18] for instance. In particular, it ensures sufficient iterations have been performed to reach the mixing time of the Markov chain associated to $P$. See Section C.3.1.

**Communication model**  We include additional details on the communication model. Consider a lockstep communication model where each round lasts for $\tau$ units of time. Within each round, agents send/receive the messages to/from their neighbours in order to implement a single update of algorithm (3). With a gradient evaluation costing 1 unit of time, each iteration of Distributed Gradient Descent takes the following amount of time $m + \tau + \text{Deg}(P)$: $m$ gradient evaluations; $\tau$ in communication delay; $\text{Deg}(P)$ for each agent to aggregating their neighbours and own gradients, as the sum in algorithm (3) $\sum_{w \in V} P_{vw}$ has computational cost $O(\text{Deg}(P))$. The delay $\tau$ can depend on factors arising from: noisy transmission, compressing or decompressing messages and synchronizing with neighbours. One particular model for $\tau$ is studied within [50] and discussed in the following remark.

**Comparison to speed-up and communication model within [50]**  The work [50] assumes the delay $\tau$ is a linear function of the network degree and some transmit time $T_{\text{Transmit}} \geq 0$ so $\tau = T_{\text{Transmit}}\text{Deg}(P)$. In our work, for sufficiently many samples $m$, the speed-up under this model for

any network topology is of the order $\frac{nm}{m+\text{Deg}(P)T_{\text{Transmit}}}$. Meanwhile, the speed-up seen within [50] is[3] of the order $\frac{nm}{m+\text{Deg}(P)T_{\text{Transmit}}}(1-\sigma_2)$, that is, same as ours but scaled by the spectral gap of the communication matrix $P$.

## B  Proof scheme

In this section we illustrate the main scheme for the proof of Theorem 2, from which Theorem 1 follows. Section B.1 presents the error decomposition into bias, variance, and network terms. Section B.2 presents the sketch of the statistical analysis for these terms, which is given in full in Section C.

### B.1  Error decomposition

The error decomposition is based on the introduction of two auxiliary processes used to compare the iterates of Distributed Gradient Descent (3).

The first auxiliary process represents the iterates generated if agents were to know the marginal distribution $\rho_X$. Initialised at $\mu_1 = 0$, the process is defined as follows for $t \geq 1$:

$$\mu_{t+1} = \mu_t - \eta_t \int_X (\langle \mu_t, x \rangle_H - f_\rho(x))x d\rho_X(x).$$

This device has already been used in the analysis of non-parametric regression in the single-machine setting [27].

The second auxiliary process represents the iterates generated if agents were to be part of a complete graph topology and were to use the protocol given by $P = \frac{1}{n}\mathbf{1}\mathbf{1}^\top$. Initialised at $\xi_{1,v} = 0$ for all $v \in V$, the process is defined as follows for $t \geq 1$:

$$\xi_{t+1,v} = \sum_{w\in V}\frac{1}{n}\left(\xi_{t,w} - \eta_t\frac{1}{m}\sum_{i=1}^{m}(\langle \xi_{t,w}, x_{i,w}\rangle_H - y_{i,w})x_{i,w}\right).$$

The analysis of iterative decentralised algorithms typically builds upon the introduction of a device analogous to this one [33, 18]. Initialised at $\xi_1 = 0$, Single-Machine Gradient Descent is defined as follows for $t \geq 1$:

$$\xi_{t+1} = \xi_t - \eta_t\frac{1}{nm}\sum_{w\in V}\sum_{i=1}^{m}\left(\langle \xi_t, x_{i,w}\rangle_H - y_{i,w}\right)x_{i,w}.$$

It is easy to see that we have $\xi_{t,v} = \xi_t$ for $t \geq 1$ and $v \in V$. This allows us to produce an analysis of Distributed Gradient Descent that relies upon known results for Single-Machine Gradient Descent.

Let us introduce the linear map $\mathcal{S}_\rho : H \to L^2(H, \rho_X)$ defined by $\mathcal{S}_\rho\omega = \langle \omega, \cdot \rangle_H$. The following error decomposition holds.

**Proposition 1.** *For any $t \geq 1$ and $v \in V$ we have*

$$\mathcal{E}(\omega_{t,v}) - \inf_{\omega\in H}\mathcal{E}(\omega) \leq 2\underbrace{\|\mathcal{S}_\rho\mu_t - f_H\|_\rho^2}_{(Bias)^2} + 4\underbrace{\|\mathcal{S}_\rho(\xi_t - \mu_t)\|_\rho^2}_{Sample\ Variance} + 4\underbrace{\|\mathcal{S}_\rho(\omega_{t,v} - \xi_{t,v})\|_\rho^2}_{Network\ Error}.$$

*Proof.* From the work in [40], $\mathcal{E}(\omega) - \inf_{\omega\in H}\mathcal{E}(\omega) = \|\mathcal{S}_\rho\omega - f_H\|_\rho^2$ for any $\omega \in H$. Adding and subtracting $\mathcal{S}_\rho\mu_t$ and using $\|x - y\|_\rho^2 \leq (\|x\|_\rho + \|y\|_\rho)^2 \leq 2\|x\|_\rho^2 + 2\|y\|_\rho^2$ we get

$$\mathcal{E}(\omega_{t,v}) - \inf_{\omega\in H}\mathcal{E}(\omega) = \|\mathcal{S}_\rho\omega_{t,v} - \mathcal{S}_\rho\mu_t + \mathcal{S}_\rho\mu_t - f_H\|_\rho^2 \leq 2\|\mathcal{S}_\rho\omega_{t,v} - \mathcal{S}_\rho\mu_t\|_\rho^2 + 2\|\mathcal{S}_\rho\mu_t - f_H\|_\rho^2.$$

Following the same steps, adding and subtracting $\mathcal{S}_\rho\xi_{t,v}$, we find

$$\|\mathcal{S}_\rho\omega_{t,v} - \mathcal{S}_\rho\mu_t\|_\rho^2 = \|\mathcal{S}_\rho\omega_{t,v} - \mathcal{S}_\rho\xi_{t,v} + \mathcal{S}_\rho\xi_{t,v} - \mathcal{S}_\rho\mu_t\|_\rho^2 \leq 2\|\mathcal{S}_\rho(\omega_{t,v} - \xi_{t,v})\|_\rho^2 + 2\|\mathcal{S}_\rho(\xi_t - \mu_t)\|_\rho^2$$

where we used the equality of $\{\xi_{s,v}\}_{s\geq 1}$ and $\{\xi_s\}_{s\geq 1}$. $\square$

Proposition 1 decomposes the error into three terms. The first term $\|\mathcal{S}_\rho \mu_t - f_H\|_\rho^2$ is deterministic and corresponds to the square of the **Bias** in the single-machine setting [57]. The second term $\|\mathcal{S}_\rho(\xi_t - \mu_t)\|_\rho^2$ aligns with what is called the **Sample Variance** in the single-machine setting, and in this case matches the sample variance obtained for Single-Machine Gradient Descent run on all $nm$ observations. The third term $\|\mathcal{S}_\rho(\omega_{t,v} - \xi_{t,v})\|_\rho^2$ accounts for the error due to performing a decentralised protocol and we call it the **Network Error**.

## B.2 Statistical analysis of error terms

In this section we illustrate the main ideas of the statistical analysis used to control the error terms in Proposition 1. Full details are given in Section C.

**Notation** Let $t$ and $k$ be positive natural numbers with $t - 1 \geq k \geq 1$. For any operator $\mathcal{L} : H \to H$, define $\Pi_{t:k+1}(\mathcal{L}) := (I - \eta_t \mathcal{L})(I - \eta_{t-1}\mathcal{L})\cdots(I - \eta_{k+1}\mathcal{L})$, with the convention $\Pi_{t:t+1}(\mathcal{L}) := I$, where $I$ is the identity operator on $H$. Let $w_{t:k+1} \equiv w_t w_{t-1} \ldots w_{k+1} := (w_t, w_{t-1}, \ldots, w_{k+1}) \in V^{t-k}$ denote a sequence of nodes in $V$. For a family of operators indexed by the nodes on the graph $\{\mathcal{L}_v\}_{v \in V}$, define $\mathcal{L}_{w_{t:k+1}} := (\mathcal{L}_{w_t}, \ldots, \mathcal{L}_{w_{k+1}})$ and $\Pi_{t:k+1}(\mathcal{L}_{w_{t:k+1}}) := (I - \eta_t \mathcal{L}_{w_t})(I - \eta_{t-1}\mathcal{L}_{w_{t-1}})\cdots(I - \eta_{k+1}\mathcal{L}_{w_{k+1}})$, with $\Pi_{t:t+1}(\mathcal{L}_{w_{t:t+1}}) := I$. Let $P_{w_{t:k+1}} := P_{w_t w_{t-1}} P_{w_{t-1} w_{t-2}} \cdots P_{w_{k+2} w_{k+1}}$ be the probability of the path generated by a Markov Chain with transition kernel $P$. For each agent $v \in V$, let $\mathcal{T}_{\mathbf{x}_v} : H \to H$ with $\mathcal{T}_{\mathbf{x}_v} = \frac{1}{m} \sum_{i=1}^m \langle \cdot, x_{i,v} \rangle_H x_{i,v}$ be the empirical covariance operator associated to the agent's own data $\mathbf{x}_v$, and let $\mathcal{T}_{\mathbf{x}_{w_{t:k+1}}} := (\mathcal{T}_{\mathbf{x}_{w_t}}, \ldots, \mathcal{T}_{\mathbf{x}_{w_{k+1}}})$. For $k \geq 1, v \in V$, let $N_{k,v} \in H$ be a random variable that only depends on the randomness in $\mathbf{z}_v$ and that has zero mean, $\mathbf{E}[N_{k,v}] = 0$. The random variable $N_{k,v}$, formally defined in (8) in Section C.3, captures the sampling error introduced at iteration $k$ of gradient descent by agent $v$. For the discussion below it suffices to mentioned the two above properties.

The following paragraphs discuss the analysis for each of the error terms.

**Bias** The analysis follows the single-machine setting and is given in Proposition 2 in Section C.1.

**Sample Variance** The analysis follows the single-machine setting [27], although the original result yields a high probability bound with a requirement on the number of samples $nm$. We therefore follow the result in [26] which yields a bound in high probability without a condition on the sample size. The bound for this term is presented in Theorem 3 in Section C.2.

**Network Error** Unraveling the iterates (Lemma 5 in Section C.3) we get, for any $v \in V, t \geq 1$:

$$\|\mathcal{S}_\rho(\omega_{t+1,v} - \xi_{t+1,v})\|_\rho = \left\| \sum_{k=1}^t \eta_k \sum_{w_{t:k} \in V^{t-k+1}} \left( P_{vw_{t:k}} - \frac{1}{n^{t-k+1}} \right) \mathcal{T}_\rho^{1/2} \Pi_{t:k+1}(\mathcal{T}_{\mathbf{x}_{w_{t:k+1}}}) N_{k,w_k} \right\|_H .$$

This characterisation makes explicit the dependence of the network error on both the communication protocol used by the agents, via the dependence on the mixing properties of the gossip matrix $P$ along each path $vw_{t:k}$, and on the statistical properties of the problem, via the product of empirical covariance operators held by the agents along each path $w_{t:k+1}$. As the randomness in the quantities $N_{k,w_k}$ might depend on the randomness in the empirical covariance operators, we further decompose the network error into two terms so that we can use the property $\mathbf{E}[N_{k,w_k}] = 0$. By adding and subtracting the terms $\Pi_{t:k+1}(\mathcal{T}_\rho)$ inside the sums we have

$$\|\mathcal{S}_\rho(\omega_{t+1,v} - \xi_{t+1,v})\|_\rho^2 \leq 2 \underbrace{\left\| \sum_{k=1}^t \eta_k \sum_{w_{t:k} \in V^{t-k+1}} \left( P_{vw_{t:k}} - \frac{1}{n^{t-k+1}} \right) \mathcal{T}_\rho^{1/2} \Pi_{t:k+1}(\mathcal{T}_\rho) N_{k,w_k} \right\|_H^2}_{(\textbf{Population Covariance Error})^2}$$

$$+ 2 \underbrace{\left\| \sum_{k=1}^t \eta_k \sum_{w_{t:k} \in V^{t-k+1}} \left( P_{vw_{t:k}} - \frac{1}{n^{t-k+1}} \right) \mathcal{T}_\rho^{1/2} \left( \Pi_{t:k+1}(\mathcal{T}_{\mathbf{x}_{w_{t:k+1}}}) - \Pi_{t:k+1}(\mathcal{T}_\rho) \right) N_{k,w_k} \right\|_H^2}_{(\textbf{Residual Empirical Covariance Error})^2} .$$

From a statistical point of view, the **Population Covariance Error** term only depends on the population covariance via the quantities $\Pi_{t:k+1}(\mathcal{T}_\rho)$, and the only source of randomness is given by $N_{k,w_k}$. Using concentration for $N_{k,w_k}$, the square of this error term can be bounded by a quantity that decreases as $\widetilde{O}(1/m)$, as announced in Section 4 alongside the discussion of Theorem 2. On the other hand, the **Residual Empirical Covariance Error** term depends on *deviations* between the empirical covariance and the population covariance via the quantities $\Pi_{t:k+1}(\mathcal{T}_{\mathbf{x}_{w_{t:k+1}}}) - \Pi_{t:k+1}(\mathcal{T}_\rho)$. Exploiting the additional concentration of these factors allows us to bound the square of this error term by a higher-order quantity that decreases as $\widetilde{O}(1/m^2)$.

We now present a separate discussion on the analysis for these two error terms, emphasizing the interplay between network topology (mixing of random walks on graphs) and statistics (concentration). The final bound for the network error is presented in Theorem 4 in Section C.3.

**Population Covariance Error** Expanding the square yields a summation over all pairs of paths:

$$\left\| \sum_{k=1}^{t} \sum_{w_{t:k} \in V^{t-k+1}} a_{k,w_{t:k}} \right\|_H^2 = \sum_{k,k'=1}^{t} \sum_{w_{t:k} \in V^{t-k+1}} \sum_{w'_{t:k'} \in V^{t-k'+1}} \langle a_{k,w_{t:k}} a_{k',w'_{t:k'}} \rangle_H$$

for properly defined quantities $a_{k,w_{t:k}}$ (the dependence on $v$ is neglected). When taking the expectation, as the random variables $\{N_{k,v}\}_{k \geq 1, v \in V}$ have zero mean and are independent across agents $v \in V$, the only paths left are those that intersect at the final node, i.e. $w_{t:k}, w'_{t:k'}$ such that $w_k = w_{k'}$. Moreover, as all agents have identically distributed data, the remaining expectation no longer depends on the final node of the paths. The remaining quantity is then analysed by bounding the probability of the two paths intersecting at the final node in terms of the second largest eigenvalue in magnitude of $P$ and by bounding the inner product by the norm product. This yields

$$\mathbf{E}[(\textbf{Pop. Cov. Error})^2] \leq \mathbf{E}\left[ \left( \sum_{k=1}^{t} \sigma_2^{t-k+1} \eta_k \| \mathcal{T}_\rho^{1/2} \Pi_{t:k+1}(\mathcal{T}_\rho) N_{k,v} \|_H \right)^2 \right].$$

Denoting the mixing time associated to $P$ as $t^\star$, the series is divided into well-mixed and poorly-mixed terms, respectively, $k \leq t - t^\star$ and $k \geq t - t^\star$. The well-mixed terms are controlled by $\sigma_2^{t-k+1}$. Meanwhile, for the poorly-mixed terms begin by taking for $\lambda > 0$ $\max_{k=1,\ldots,t} \{ \|(\mathcal{T}_\rho + \lambda I)^{-1/2} N_{k,v}\|_H^2 \}$ outside of the series. The expectation of this maximum is controlled through concentration and becomes $\widetilde{O}(\frac{1}{m^2\lambda} + \frac{1}{m\lambda^{\gamma'}})$ for $\gamma' \in [1, \gamma]$. The remaining series is controlled through the contraction of the term $\| \mathcal{T}_\rho^{1/2} \Pi_{t:k+1}(\mathcal{T}_\rho)(\mathcal{T}_\rho + \lambda I)^{1/2} \|$ and choosing $\lambda \simeq 1/(\eta t^\star)$. These two steps lead to this term being of the order $O(\frac{\eta t^\star}{m^2} + \frac{(\eta t^\star)^{\gamma'}}{m})$, which dominates the well-mixed terms and contributes to the dependence on the inverse of the spectral gap of $P$. The free parameter $\gamma' \in [1, \gamma]$ is left open as a smaller step size $\eta$ is used to control this term when $m \leq n^{2r/\gamma}$. The final bound is given in Lemma 8 in Section C.3.1.

**Residual Empirical Covariance Error** The analysis of this term is based on the following identity (Proposition 5 in Section C.3.2), for any $t - 1 \geq k$ and any $w_{t:k+1} \in V^{t-k}$:

$$\Pi_{t:k+1}(\mathcal{T}_{\mathbf{x}_{w_{t:k+1}}}) - \Pi_{t:k+1}(\mathcal{T}_\rho) = \sum_{j=k+1}^{t} \eta_j \Pi_{t:j+1}(\mathcal{T}_\rho)(\mathcal{T}_\rho - \mathcal{T}_{\mathbf{x}_{w_j}})\Pi_{j-1:k+1}(\mathcal{T}_{\mathbf{x}_{w_{j-1:k+1}}}).$$

The above decomposition has two key properties. Firstly, it depends upon differences between the empirical covariance operators $\mathcal{T}_{\mathbf{x}_{w_j}}$ and its expectation $\mathcal{T}_\rho$. This allows concentration to be used, and, alongside the concentration for $N_{k,v}$, it ensures that $(\textbf{Resid. Emp. Cov. Error})^2$ is of order $\widetilde{O}(1/m^2)$. Secondly, it is of the form $\sum_{j=k+1}^{t} \eta_j \Pi_{t:j+1}(\mathcal{T}_\rho)[\cdots]$, where $[\cdots]$ indicates the right most factors and the quantity shown aligns with the filter function for gradient descent [26, Example 2]. Once again the contractive property of the quantity $\Pi_{t:j+1}(\mathcal{T}_\rho)$ allows to give sharper rates with respect to the step size and number of iterations. Without it, the choice of step size $\eta_t = \eta t^{-\theta}$ would yield a bound for $(\textbf{Resid. Emp. Cov. Error})^2$ of the order $\left( \sum_{k=1}^{t} \eta_k \sum_{j=k+1}^{t-1} \eta_j \right)^2 \simeq (\eta t^{1-\theta})^4$. The contraction allows to show that $(\textbf{Resid. Emp. Cov. Error})^2$ grows at the reduced order $(\eta t^{1-\theta})^3$,

and the addition of the capacity assumption allows it to be further reduced to the order $(\eta t^{1-\theta})^{2+\gamma}$. The final high-probability bound is given in Lemma 9 in Section C.3.2. This being stronger than the bound in expectation required for Theorem 2.

## C  Proofs

Before going on to present proofs for the main result some notation is introduced following [40, 27]. Some notation is repeated from the previous sections, as additional details are included. Adopt the convention for sums $\sum_{k=t+1}^{t} = 0$. For a given bounded operator $\mathcal{L} : L^2(H, \rho_X) \to H$, let $\|\mathcal{L}\|$ denote the operator norm of $\mathcal{L}$, i.e. $\|\mathcal{L}\| = \sup_{f \in L^2(H,\rho_X), \|f\|_\rho=1} \|\mathcal{L}f\|_H$. Let $\mathcal{S}_\rho : H \to L^2(H, \rho_X)$ be the linear map $\omega \to \langle \omega, \cdot \rangle_H$,which is bounded by $\kappa$ under Assumption 1. Consider the adjoint operator $\mathcal{S}_\rho^\star : L^2(H, \rho_X) \to H$, the covariance operator $\mathcal{T}_\rho : H \to H$ given by $\mathcal{T}_\rho = \mathcal{S}_\rho^\star \mathcal{S}_\rho$, and the operator $\mathcal{L}_\rho : L^2(H, \rho_X) \to L^2(H, \rho_X)$ given by $\mathcal{L}_\rho = \mathcal{S}_\rho \mathcal{S}_\rho^\star$. We have $\mathcal{S}_\rho^\star g = \int_X x g(x) d\rho_X(x)$ and $\mathcal{T}_\rho = \int_X \langle \cdot, x \rangle_H x d\rho_X(x)$. For any $\omega \in H$ the following isometry property holds [48]

$$\|\mathcal{S}_\rho \omega\|_\rho = \|\sqrt{\mathcal{T}_\rho} \omega\|_H.$$

The following notation was utilised in the analysis of Single-Machine Gradient Descent [40, 27]. In this case it aligns with all of the observations in the network $\mathbf{y} := \{y_{i,v}\}_{i=1,\ldots,m,v\in V} \in \mathbb{R}^{m|V|}$ and $\mathbf{x} = \{x_{i,v}\}_{i=1,\ldots,m,v\in V}$. Define the sampling operator $\mathcal{S}_\mathbf{x} : H \to \mathbb{R}^{m|V|}$ by $(\mathcal{S}_\mathbf{x}\omega)_{(i,v)} = \langle \omega, x_{i,v} \rangle_H$, for $i = 1, \ldots, m, v \in V$. Let $\| \cdot \|_{\mathbb{R}^{m|V|}}$ denote the Euclidean norm in in $\mathbb{R}^{m|V|}$ times the factor $1/\sqrt{nm}$. Its adjoint operator $\mathcal{S}_\mathbf{x}^\star : \mathbb{R}^{m|V|} \to H$, defined by $\langle \mathcal{S}_\mathbf{x}^\star \mathbf{y}, \omega \rangle_H = \langle \mathbf{y}, \mathcal{S}_\mathbf{x}\omega \rangle_{\mathbb{R}^{m|V|}}$ for $\mathbf{y} \in \mathbb{R}^{m|V|}$, is given by $\mathcal{S}_\mathbf{x}^\star \mathbf{y} = \frac{1}{nm} \sum_{v\in V} \sum_{i=1}^{m} y_{i,v} x_{i,v}$. Define the covariance operator with respect to all of the samples $\mathcal{T}_\mathbf{x} : H \to H$ such that $\mathcal{T}_\mathbf{x} = \mathcal{S}_\mathbf{x}^\star \mathcal{S}_\mathbf{x}$. We have

$$\mathcal{T}_\mathbf{x} = \frac{1}{nm} \sum_{v\in V} \sum_{i=1}^{m} \langle \cdot, x_{i,v} \rangle_H x_{i,v}.$$

The following notation is analogous to the single-machine notation just introduced, although now with respect to the datasets held by individual agents, i.e. $\mathbf{x}_v$ and $\mathbf{y}_v$ for $v \in V$. Let $\mathcal{S}_{\mathbf{x}_v} : H \to \mathbb{R}^m$ with $(\mathcal{S}_{\mathbf{x}_v}\omega)_i = \langle \omega, x_{i,v} \rangle_H$ for $i = 1, \ldots, m$. Let $\| \cdot \|_{\mathbb{R}^m}$ be the Euclidean norm in $\| \cdot \|_{\mathbb{R}^m}$ times $1/\sqrt{m}$. Its adjoint operator $\mathcal{S}_{\mathbf{x}_v}^\star : \mathbb{R}^m \to H$, defined by $\langle \mathcal{S}_{\mathbf{x}_v}^\star \mathbf{y}_v, \omega \rangle_H = \langle \mathbf{y}_v, \mathcal{S}_{\mathbf{x}_v}\omega \rangle_{\mathbb{R}^m}$ for $\mathbf{y}_v \in \mathbb{R}^m$, is given by $\mathcal{S}_{\mathbf{x}_v}^\star \mathbf{y}_v = \frac{1}{m} \sum_{i=1}^{m} y_{i,v} x_{i,v}$. The empirical covariance operator $\mathcal{T}_{\mathbf{x}_v} : H \to H$ is such that $\mathcal{T}_{\mathbf{x}_v} = \mathcal{S}_{\mathbf{x}_v}^\star \mathcal{S}_{\mathbf{x}_v}$, with $\mathcal{T}_{\mathbf{x}_v} = \frac{1}{m} \sum_{i=1}^{m} \langle \cdot, x_{i,v} \rangle_H x_{i,v}$.

Using this notation, the processes $\{\mu_t\}_{t\geq 1}$, $\{\omega_{t,v}\}_{t\geq 1}$, and $\{\xi_t\}_{t\geq 1}$ can be rewritten as follows. The population process reads

$$\mu_{t+1} = \mu_t - \eta_t (\mathcal{T}_\rho \mu_t - \mathcal{S}_\rho^\star f_\rho).$$

The gossiped process reads

$$\omega_{t+1,v} = \sum_{w\in V} P_{vw} \Big( \omega_{t,w} - \eta_t (\mathcal{T}_{\mathbf{x}_w} \omega_{t,w} - \mathcal{S}_{\mathbf{x}_w}^\star \mathbf{y}_w) \Big).$$

The single-machine process reads

$$\xi_{t+1} = \xi_t - \eta_t (\mathcal{T}_\mathbf{x} \xi_t - \mathcal{S}_\mathbf{x}^\star \mathbf{y}).$$

The next three sections present bounds for the three error terms introduced in Proposition 1. Section C.1 presents a bound for the Bias term, which follows directly from the results in [27] and references therein. Section C.2 establishes a bound for the Sample Variance term, which follows from results in [26]. Section C.3 develops bounds for the Network Error term, which are a novel contribution of this work. Section C.4 brings the results of the previous three sections together to establish the proofs of Theorem 2 and Theorem 1, respectively. Section C.5 includes useful inequalities that are needed to establish our results.

### C.1  Bias

The following bound on the Bias term $\|\mathcal{S}_\rho \mu_t - f_H\|_\rho^2$ is taken from [27], inspired by [57, 40].

**Proposition 2.** *[27, Appendix C Proposition 2] Under Assumption 2, let $\eta\kappa^2 \leq 1$. Then for any $t \in \mathbb{N}$,*

$$\|\mathcal{S}_\rho\mu_t - f_H\|_\rho \leq R\left(\frac{r}{2\sum_{j=1}^t \eta_j}\right)^r.$$

*In particular, if $\eta_t = \eta t^{-\theta}$ for all $t \in \mathbb{N}$, with $\eta \in (0, \kappa^{-2}]$ and $\theta \in [0, 1)$ then*

$$\|\mathcal{S}_\rho\mu_t - f_H\|_\rho \leq Rr^r\eta^{-r}t^{r(\theta-1)}.$$

## C.2 Sample Variance

In this section we establish a bound for the expectation of the Sample Variance term $\mathbf{E}[\|\mathcal{S}_\rho(\xi_t - \mu_t)\|_\rho^2]$. The following lemma summaries a number of intermediary steps in [27] for bounding the Sample Variance term. It arises from representing the iterates $\{\xi_t - \mu_t\}_{t \geq 1}$ in terms of the stochastic sequence $\{N_k\}_{k \geq 1}$ which characterises the sample noise introduced in the iterations of gradient descent. These terms are controlled via the empirical covariance operator $\mathcal{T}_\mathbf{x}$ and the population covariance operator $\mathcal{T}_\rho$ while introducing the pseudo-regularisation parameter $\lambda > 0$ and utilising the contractive property of the gradient updates. For the following, let us introduce the notation $\mathcal{T}_{\rho,\lambda} = \mathcal{T}_\rho + \lambda I$ and $\mathcal{T}_{\mathbf{x},\lambda} = \mathcal{T}_\mathbf{x} + \lambda I$.

**Lemma 1.** *Let $\eta_1\kappa^2 \leq 1$ and $0 \leq \lambda$. For any $t \in \mathbb{N}$ we have*

$$\|\mathcal{S}_\rho(\xi_{t+1} - \mu_{t+1})\|_\rho$$
$$\leq \left(\sum_{k=1}^{t-1} \frac{\eta_k\|\mathcal{T}_{\rho,\lambda}^{-1/2}N_k\|_H}{2\sum_{i=k+1}^t \eta_i} + \lambda\sum_{k=1}^{t-1}\eta_k\|\mathcal{T}_{\rho,\lambda}^{-1/2}N_k\|_H + \|\mathcal{T}_\rho\|^{1/2}(\|\mathcal{T}_\rho\| + \lambda)^{1/2}\eta_t\|\mathcal{T}_{\rho,\lambda}^{-1/2}N_t\|_H\right)$$
$$\times \|\mathcal{T}_{\mathbf{x},\lambda}^{-1/2}\mathcal{T}_\rho^{1/2}\|\|\mathcal{T}_{\mathbf{x},\lambda}^{-1/2}\mathcal{T}_{\rho,\lambda}^{1/2}\|,$$

*where*

$$N_k = (\mathcal{T}_\rho\mu_k - \mathcal{S}_\rho^\star f_\rho) - (\mathcal{T}_\mathbf{x}\mu_k - \mathcal{S}_\mathbf{x}^\star\mathbf{y}), \quad \forall k \in \mathbb{N}. \tag{7}$$

*Proof.* The proof of this result follows the proof of [27, Proposition 3]. $\square$

The two quantities left to control are $\|\mathcal{T}_{\rho,\lambda}^{-1/2}N_k\|_H$ for $k \in \mathbb{N}$ as well as $\|(\mathcal{T}_\mathbf{x} + \lambda I)^{-1/2}\mathcal{T}_\rho^{1/2}\|^2$. The first of these quantities is controlled by [27, Lemma 18] which is summarised in the following lemma.

**Lemma 2.** *[27, Lemma 18] Let Assumptions 1, 2, 3 hold with $r \geq 1/2$ and $\{N_k\}_{k \geq 1}$ be as in (7). For any $\lambda > 0$, with probability at least $1 - \delta$, the following holds $\forall k \in \mathbb{N}$*

$$\|(\mathcal{T}_\rho + \lambda I)^{-1/2}N_k\|_H \leq 4(R\kappa^{2r} + \sqrt{M})\left(\frac{\kappa}{nm\sqrt{\lambda}} + \frac{\sqrt{2\sqrt{\nu}c_\gamma}}{\sqrt{nm\lambda^\gamma}}\right)\log\frac{4}{\delta}.$$

The next lemma from [26, Lemma 19 Remark 1] controls $\|(\mathcal{T}_\mathbf{x} + \lambda I)^{-1/2}\mathcal{T}_\rho^{1/2}\|^2$.

**Lemma 3.** *[26, Lemma 19, Remark 1] Let $\delta \in (0, 1)$ and $\lambda = (nm)^{-p}$ for some $p \geq 0$. With probability at least $1 - \delta$ the following holds*

$$\|\mathcal{T}_\rho^{1/2}(\mathcal{T}_\mathbf{x} + \lambda)^{-1/2}\|^2 \leq \|(\mathcal{T}_\rho + \lambda I)^{1/2}(\mathcal{T}_\mathbf{x} + \lambda)^{-1/2}\|^2$$
$$\leq 24\kappa^2\left(\log\frac{4\kappa^2(c_\gamma + 1)}{\delta\|\mathcal{T}_\rho\|} + p\gamma\min\left(\frac{1}{e(1-p)_+}, \log nm\right)\right)(1 \vee (nm)^{p-1}).$$

Bringing together the three previous results yields the following high-probability bound for the Sample Variance term.

**Proposition 3.** *Fix $\delta \in (0, 1)$ and $p \in (0, 1)$. Let Assumptions 1, 2 and 3 hold with $r \geq 1/2$ and $\eta_t = \eta t^{-\theta}$ with $\eta\kappa^2 \leq 1$, $\theta \in [0, 1)$. The following holds with probability at least $1 - \delta$ for any $t \in \mathbb{N}$*

$$\|\mathcal{S}_\rho(\xi_{t+1} - \mu_{t+1})\|_\rho$$
$$\leq \widetilde{d}_1\min\left(\frac{1}{e(1-p)_+}, \log nm\right)\frac{\log(t)}{(nm)^{(1-p\gamma)/2}}(1 \vee (nm)^{-p}\eta t^{1-\theta} \vee \eta t^{-\theta})\log^2\frac{\widetilde{d}_2}{\delta},$$

*with $\widetilde{d}_1 = 768\frac{\kappa^2\|\mathcal{T}_\rho\|^{1/2}(\|\mathcal{T}_\rho\|+1)^{1/2}(R\kappa^{2r}+\sqrt{M})(\kappa+\sqrt{2\sqrt{\nu}c_\gamma})}{1-\theta}$ and $\widetilde{d}_2 = 8\left(1 \vee \kappa^2\frac{(c_\gamma+1)}{\|\mathcal{T}_\rho\|}\right)$.*

*Proof.* Fix $\delta \in (0,1)$ and set $\lambda = (nm)^{-p}$ with $p \in (0,1)$. Lemma 2 implies that with probability at least $1 - \frac{\delta}{2}$ the following holds for any $k \in \mathbb{N}$

$$\|(\mathcal{T}_\rho + \lambda I)^{-1/2} N_k\|_H \le 4(R\kappa^{2r} + \sqrt{M})\left(\kappa + \sqrt{2\sqrt{\nu}c_\gamma}\right)\frac{\log\frac{8}{\delta}}{(nm)^{(1-p\gamma)/2}}.$$

Similarly, Lemma 3 implies that the following holds with probability at least $1 - \frac{\delta}{2}$

$$\|\mathcal{T}_\rho^{1/2}(\mathcal{T}_\mathbf{x} + \lambda I)^{-1/2}\|^2 \le \|\mathcal{T}_{\rho,\lambda}^{1/2}(\mathcal{T}_\mathbf{x} + \lambda I)^{-1/2}\|^2$$

$$\le 48\kappa^2 \min\left(\frac{1}{e(1-p)_+}, \log nm\right) \log \frac{8\kappa^2(c_\gamma + 1)}{\delta\|\mathcal{T}_\rho\|}.$$

Following [27], the series can be bounded as follows

$$\sum_{k=1}^{t-1} \frac{\eta_k}{2\sum_{i=k+1}^t \eta_i} + \lambda \sum_{k=1}^{t-1} \eta_k + \|\mathcal{T}_\rho\|^{1/2}(\|\mathcal{T}_\rho\| + \lambda)^{1/2}\eta_t$$

$$\le 2\log(t) + \frac{\lambda\eta t^{1-\theta}}{1-\theta} + \|\mathcal{T}_\rho\|^{1/2}(\|\mathcal{T}_\rho\| + 1)^{1/2}\eta t^{-\theta}$$

$$\le \frac{4\|\mathcal{T}_\rho\|^{1/2}(\|\mathcal{T}_\rho\| + 1)^{1/2}\log(t)}{1-\theta}(1 \vee (\lambda\eta t^{1-\theta}) \vee (\eta t^{-\theta})),$$

where we used $\lambda = (nm)^{-p} \le 1$ to get $(\|\mathcal{T}_\rho\| + \lambda)^{1/2} \le (\|\mathcal{T}_\rho\| + 1)^{1/2}$. Plugging everything into Lemma 1 and using a union bound we obtain that the result holds with probability at least $1 - \frac{\delta}{2} - \frac{\delta}{2} = 1 - \delta$. $\qquad\square$

Proposition 3 gives a bound that holds with high probability. We make use of the following lemma to derive a bound in expectation.

**Lemma 4.** *[7, Appendix Lemma C.1] Let $F : (0,1] \to \mathbb{R}_+$ be a monotone, non-increasing, continuous function and $V$ a non-negative real-valued random variable such that*

$$\mathbf{P}[V > F(t)] \le t, \quad \forall t \in (0,1].$$

*Then we have $\mathbf{E}[V] \le \int_0^1 F(t)dt$.*

The following theorem presents the final bound for the expected value of the Sample Variance term.

**Theorem 3.** *Let Assumptions 1, 2, 3 hold with $r \ge 1/2$, $p \in (0,1)$ and $\eta_t = \eta t^{-\theta}$ for all $t \in \mathbb{N}$ with $\eta \in (0, \kappa^{-2}]$, $\theta \in [0,1)$. Then for following holds for all $t \in \mathbb{N}$:*

$$\mathbf{E}[\|\mathcal{S}_\rho(\xi_t - \mu_t)\|_\rho^2]$$

$$\le \widetilde{d}_3 \min\left(\frac{1}{e(1-p)_+}, \log nm\right)^2 \frac{\log^2(t)}{(nm)^{(1-p\gamma)}}\left(1 \vee ((nm)^{-p}\eta t^{1-\theta})^2 \vee t^{-2}(\eta t^{1-\theta})^2\right),$$

*with $\widetilde{d}_3 = 64\widetilde{d}_1^2 \log^4 \widetilde{d}_2$ and with $\widetilde{d}_1$, $\widetilde{d}_2$ defined as in Proposition 3.*

*Proof.* Consider the term $\|\mathcal{S}_\rho(\xi_t - \mu_t)\|_\rho^2$. Utilising the high-probability bound in Proposition 3 as well as Lemma 4, the expectation of the squared norm can be bounded as

$$\mathbf{E}[\|\mathcal{S}_\rho(\xi_t - \mu_t)\|_\rho^2]$$

$$\le \widetilde{d}_1^2 \min\left(\frac{1}{e(1-p)_+}, \log nm\right)^2 \frac{\log^2(t)}{(nm)^{(1-p\gamma)}}\left(1 \vee ((nm)^{-p}\eta t^{1-\theta})^2 \vee t^{-2}(\eta t^{1-\theta})^2\right)$$

$$\times \int_0^1 \log^4 \frac{\widetilde{d}_2}{\delta} d\delta.$$

The result follows by using the bound $\int_0^1 \log^4 \frac{\widetilde{d}_2}{\delta} d\delta \le 64 \log^4(\widetilde{d}_2)$. $\qquad\square$

## C.3 Network Error

In this section we develop the bound for the Network Error term. The following lemma shows that the error can be decomposed into terms similar to $\{N_k\}_{k\in\mathbb{N}}$ defined in (7) for the Sample Variance.

**Lemma 5.** *For all $t \in \mathbb{N}$ we have*

$$\|\mathcal{S}_\rho(\omega_{t+1,v} - \xi_{t+1,v})\|_\rho = \left\| \sum_{k=1}^{t} \eta_k \sum_{w_{t:k} \in V^{t-k+1}} \left(P_{vw_{t:k}} - \frac{1}{n^{t-k+1}}\right) \mathcal{T}_\rho^{1/2} \Pi_{t:k+1}(\mathcal{T}_{\mathbf{x}_{w_{t:k+1}}}) N_{k,w_k} \right\|_H,$$

*where*

$$N_{k,v} := (\mathcal{T}_\rho \mu_k - \mathcal{S}_\rho^\star f_\rho) - (\mathcal{T}_{\mathbf{x}_v} \mu_k - \mathcal{S}_{\mathbf{x}_v}^\star \mathbf{y}_v), \quad \forall k \in \mathbb{N}, \ v \in V. \tag{8}$$

*Proof.* For $t \geq 1$ the difference between the iterates $\omega_{t+1,v} - \mu_{t+1}$ can be written as follows

$$\omega_{t+1,v} - \mu_{t+1} = \sum_{w \in V} P_{vw}\left(\omega_{t,w} - \mu_t + \eta_t\{(\mathcal{T}_\rho\mu_t - \mathcal{S}_\rho^\star f_\rho) - (\mathcal{T}_{\mathbf{x}_w}\omega_{t,w} - \mathcal{S}_{\mathbf{x}_w}^\star \mathbf{y}_w)\}\right)$$

$$= \sum_{w \in V} P_{vw}\left((I - \eta_t \mathcal{T}_{\mathbf{x}_w})(\omega_{t,w} - \mu_t) + \eta_t \underbrace{\{(\mathcal{T}_\rho\mu_t - \mathcal{S}_\rho^\star f_\rho) - (\mathcal{T}_{\mathbf{x}_w}\mu_t - \mathcal{S}_{\mathbf{x}_w}^\star \mathbf{y}_w)\}}_{N_{t,w}}\right)$$

$$= \sum_{w \in V} P_{vw}\left((I - \eta_t \mathcal{T}_{\mathbf{x}_w})(\omega_{t,w} - \mu_t) + \eta_t N_{t,w}\right).$$

Unravelling the iterates and using $\omega_1 = \mu_1 = 0$ yield

$$\omega_{t+1,v} - \mu_{t+1} = \sum_{w_{t:1} \in V^t} P_{vw_{t:1}} \Pi_{t:1}(\mathcal{T}_{\mathbf{x}_{w_{t:1}}})(\omega_1 - \mu_1) + \sum_{k=1}^{t} \eta_k \sum_{w_{t:k} \in V^t} P_{vw_{t:k}} \Pi_{t:k+1}(\mathcal{T}_{\mathbf{x}_{w_{t:k+1}}}) N_{k,w_k}$$

$$= \sum_{k=1}^{t} \eta_k \sum_{w_{t:k} \in V^{t-k+1}} P_{vw_{t:k}} \Pi_{t:k+1}(\mathcal{T}_{\mathbf{x}_{w_{t:k+1}}}) N_{k,w_k}.$$

The iterates $\xi_{t+1,v} - \mu_{t+1}$ are similarly written and unravelled using $\xi_{1,v} = 0$:

$$\xi_{t+1,v} - \mu_{t+1} = \sum_{w \in V} \frac{1}{n}\left((I - \eta_t \mathcal{T}_{\mathbf{x}_w})(\xi_{t,w} - \mu_t) + \eta_t N_{t,w}\right)$$

$$= \sum_{k=1}^{t} \eta_k \sum_{w_{t:k} \in V^{t-k+1}} \frac{1}{n^{t-k+1}} \Pi_{t:k+1}(\mathcal{T}_{\mathbf{x}_{w_{t:k+1}}}) N_{k,w_k}.$$

The deviation $\omega_{t+1} - \xi_{t+1,v}$ can then be written as follows

$$\omega_{t+1,v} - \xi_{t+1,v} = \sum_{k=1}^{t} \eta_k \sum_{w_{t:k} \in V^{t-k+1}} \left(P_{vw_{t:k}} - \frac{1}{n^{t-k+1}}\right) \Pi_{t:k+1}(\mathcal{T}_{\mathbf{x}_{w_{t:k+1}}}) N_{k,w_k}.$$

Applying $\mathcal{S}_\rho$, taking norm $\|\cdot\|_\rho$ on both sides and using the isometry property yields the result. $\quad\square$

For $v, w \in V$ and $k \geq 1$, we want to exploit that the random variables $N_{k,v}$ and $N_{k,w}$ have zero mean, $\mathbf{E}[N_{k,v}] = 0$, and are independent for $v \neq w$. To do so we add and subtract $\Pi_{t:k+1}(\mathcal{T}_\rho)$ inside the norm so the following upper bound can be formed:

$$\|\mathcal{S}_\rho(\omega_{t+1,v} - \xi_{t+1,v})\|_\rho^2 \tag{9}$$

$$\leq 2 \underbrace{\left\| \sum_{k=1}^{t} \eta_k \sum_{w_{t:k} \in V^{t-k+1}} \left(P_{vw_{t:k}} - \frac{1}{n^{t-k+1}}\right) \mathcal{T}_\rho^{1/2} \Pi_{t:k+1}(\mathcal{T}_\rho) N_{k,w_k} \right\|_H^2}_{(\textbf{Population Covariance Error})^2}$$

$$+ 2 \underbrace{\left\| \sum_{k=1}^{t} \eta_k \sum_{w_{t:k} \in V^{t-k+1}} \left(P_{vw_{t:k}} - \frac{1}{n^{t-k+1}}\right) \mathcal{T}_\rho^{1/2} \left(\Pi_{t:k+1}(\mathcal{T}_{\mathbf{x}_{w_{t:k+1}}}) - \Pi_{t:k+1}(\mathcal{T}_\rho)\right) N_{k,w_k} \right\|_H^2}_{(\textbf{Residual Empirical Covariance Error})^2}.$$

The **Population Covariance Error** (**Pop. Cov. Error**) will be controlled by using the independence of the terms $\{N_{k,w}\}_{w \in V}$. The **Residual Empirical Covariance Error** (**Resid. Emp. Cov. Error**) will be analysed by decomposing it into terms that concentrate to zero sufficiently quickly.

The following lemma, similar to Lemma 2 for the sample variance, gives concentration rates for the quantities held by the individual agents.

**Lemma 6.** *Fix $v \in V$. Let Assumptions 1, 2, 3 hold with $r \geq 1/2$ and $\{N_{s,v}\}_{s \in \mathbb{N}}$ be defined as in (8). For any $\lambda > 0$, with probability at least $1 - \delta$, the following holds for all $k \in \mathbb{N}$:*

$$\|(\mathcal{T}_\rho + \lambda I)^{-1/2} N_{k,v}\|_H \leq 4(R\kappa^{2r} + \sqrt{M})\left(\frac{\kappa}{m\sqrt{\lambda}} + \frac{\sqrt{2\sqrt{\nu}c_\gamma}}{\sqrt{m\lambda^\gamma}}\right) \log \frac{4}{\delta}. \qquad (10)$$

*Let $\| \cdot \|_{HS}$ denote the Hilbert-Schmidt norm of a bounded operator from $H$ to $H$. The following holds with probability at least $1 - \delta$:*

$$\|(\mathcal{T}_\rho + \lambda I)^{-1/2}(\mathcal{T}_\rho - \mathcal{T}_{\mathbf{x}_v})\|_{HS} \leq 2\kappa\left(\frac{2\kappa}{m\sqrt{\lambda}} + \frac{\sqrt{c_\gamma}}{\sqrt{m\lambda^\gamma}}\right) \log \frac{4}{\delta}. \qquad (11)$$

*Proof.* Both inequalities arise from concentration results for random variables in Hilbert spaces used in [12] and based on results in [36]. Inequalities (10,11) come directly from [27, Lemma 18], where in particular (11) was used to prove (10). □

We now move on to establish bounds for the **Population Covariance Error** term and the **Residual Empirical Covariance Error** term within the following two sections, Section C.3.1 and Section C.3.2, respectively. Section C.3.3 then brings together the previously developed results to establish a bound for the Network Error term.

We will need the following lemma, taken from [27, Lemma 15], which itself follows [58, 49].

**Lemma 7.** *Let $\mathcal{L}$ be a compact, positive operator on a separable Hilbert Space $H$. Assume that $\eta\|\mathcal{L}\| \leq 1$. For $t \in \mathbb{N}$, $a > 0$ and any non-negative integer $k \leq t - 1$ we have*

$$\|\Pi_{t:k+1}(\mathcal{L})\mathcal{L}^a\| \leq \left(\frac{a}{e\sum_{j=k+1}^t \eta_j}\right)^a.$$

*Proof.* The proof in [27, Lemma 15] considers this result with $a = r$. The proof for more general $a > 0$ follows the same steps. □

### C.3.1 Analysis of Population Covariance Error

In this section we develop a bound for the **Population Covariance Error** term in (9). The final result is presented in Lemma 8.

The following proposition bounds the expectation of (**Population Covariance Error**)$^2$ by a series involving the products of (deterministic) operators $\{\mathcal{T}_\rho^{1/2}\Pi_{t:k+1}(\mathcal{T}_\rho)\}$, as a function of the step size, the largest eigenvalue in absolute value of the gossip matrix $P$, and the random variables $\{N_{k,w}\}$.

**Proposition 4.** *For any $t \in \mathbb{N}$ and $v \in V$ we have*

$$\mathbf{E}\left[\left\|\sum_{k=1}^t \eta_k \sum_{w_{t:k} \in V^{t-k+1}} \left(P_{vw_{t:k}} - \frac{1}{n^{t-k+1}}\right)\mathcal{T}_\rho^{1/2}\Pi_{t:k+1}(\mathcal{T}_\rho)N_{k,w_k}\right\|_H^2\right]$$

$$\leq \mathbf{E}\left[\left(\sum_{k=1}^t \sigma_2^{t-k+1}\eta_k\|\mathcal{T}_\rho^{1/2}\Pi_{t:k+1}(\mathcal{T}_\rho)N_{k,v}\|_H\right)^2\right].$$

*Proof.* Fix $t \in \mathbb{N}$ and $v \in V$. Let us introduce the notation $\Delta(w_{t:k}) := \left(P_{vw_{t:k}} - \frac{1}{n^{t-k+1}}\right)$. Expanding the square and taking the expectation we get

$$\mathbf{E}\left[\left\|\sum_{k=1}^{t} \eta_k \sum_{w_{t:k} \in V^{t-k+1}} \left(P_{vw_{t:k}} - \frac{1}{n^{t-k+1}}\right) \mathcal{T}_\rho^{1/2} \Pi_{t:k+1}(\mathcal{T}_\rho) N_{k,w_k}\right\|_H^2\right]$$

$$= \sum_{k,k'=1}^{t} \eta_k \eta_{k'} \sum_{\substack{w_{t:k} \in V^{t-k+1} \\ w'_{t:k'} \in V^{t-k'+1}}} \Delta(w_{t:k}) \Delta(w'_{t:k'}) \mathbf{E}\langle \mathcal{T}_\rho^{1/2} \Pi_{t:k+1}(\mathcal{T}_\rho) N_{k,w_k}, \mathcal{T}_\rho^{1/2} \Pi_{t:k'+1}(\mathcal{T}_\rho) N_{k',w'_{k'}}\rangle_H$$

$$= \sum_{k,k'=1}^{t} \eta_k \eta_{k'} \mathbf{E}\langle \mathcal{T}_\rho^{1/2} \Pi_{t:k+1}(\mathcal{T}_\rho) N_{k,v}, \mathcal{T}_\rho^{1/2} \Pi_{t:k'+1}(\mathcal{T}_\rho) N_{k',v}\rangle_H \sum_{\substack{w_{t:k} \in V^{t-k+1} \\ w'_{t:k'} \in V^{t-k'+1} \\ w_k = w'_{k'}}} \Delta(w_{t:k}) \Delta(w'_{t:k'}).$$

The last identity follows from the fact that the samples held by agents are independent and identically distributed. As the agents' datasets are independent, the inner products are zero for $k, k' \in \{1, \ldots, t\}$ whenever the final elements of the paths $w_{t:k}$ and $w'_{t:k'}$ do not coincide, i.e.

$$\mathbf{E}\langle \mathcal{T}_\rho^{1/2} \Pi_{t:k+1}(\mathcal{T}_\rho) N_{k,w_k}, \mathcal{T}_\rho^{1/2} \Pi_{t:k'+1}(\mathcal{T}_\rho) N_{k',w'_{k'}}\rangle_H = 0 \text{ if } w_k \neq w'_{k'}.$$

As the agents' datasets are identically distributed, the expectation of the inner products can be taken outside the sum over the paths. The sum over all pairs of paths that intersect at the final node can be simplified as follows:

$$\sum_{\substack{w_{t:k} \in V^{t-k+1} \\ w'_{t:k'} \in V^{t-k'+1} \\ w_k = w'_{k'}}} \Delta(w_{t:k}) \Delta(w'_{t:k'})$$

$$= \sum_{\substack{w_k, w'_{k'} \in V \\ w_k = w'_{k'}}} \sum_{w_{t:k+1} \in V^{t-k}} \sum_{w'_{t:k'+1} \in V^{t-k'}} \left(P_{vw_{t:k}} - \frac{1}{n^{t-k+1}}\right) \left(P_{vw'_{t:k'}} - \frac{1}{n^{t-k'+1}}\right)$$

$$= \sum_{w \in V} \left((P^{t-k+1})_{vw} - \frac{1}{n}\right) \left((P^{t-k'+1})_{vw} - \frac{1}{n}\right).$$

For each $v \in V$ let $e_v \in \mathbb{R}^n$ denote the vector of all zeros but a 1 in the place aligned with agent $v$. The summation can be further simplified by utilising the assumption that $P$ is symmetric and doubly-stochastic, i.e. $P^\top = P$ and $P\mathbf{1} = \mathbf{1}$. By the eigendecomposition of the gossip matrix $P$, recall Section 2.3, for any $s > 0$ we have $(P^s)_{vv} = \sum_{l=1}^{n} \lambda_l^s u_{l,v}^2 = \frac{1}{n} + \sum_{l=2}^{n} \lambda_l^s u_{l,v}^2$. This yields the bound $|(P^s)_{vv} - \frac{1}{n}| = |\sum_{l=2}^{n} \lambda_l^s u_{l,v}^2| \leq \sigma_2^s \sum_{l=2}^{n} u_{l,v}^2 \leq \sigma_2^s$ where $\sigma_2 := \max\{|\lambda_2|, |\lambda_n|\}$ is the second largest eigenvalue in absolute value. Bringing everything together, the expected norm of (**Pop. Cov. Error**)$^2$ can be written and bounded as follows:

$$\mathbf{E}\left[\left\|\sum_{k=1}^{t} \eta_k \sum_{w_{t:k} \in V^{t-k+1}} \left(P_{vw_{t:k}} - \frac{1}{n^{t-k+1}}\right) \mathcal{T}_\rho^{1/2} \Pi_{t:k+1}(\mathcal{T}_\rho) N_{k,w_k}\right\|_H^2\right]$$

$$= \sum_{k,k'=1}^{t} \eta_k \eta_{k'} \mathbf{E}\langle \mathcal{T}_\rho^{1/2} \Pi_{t:k+1}(\mathcal{T}_\rho) N_{k,v}, \mathcal{T}_\rho^{1/2} \Pi_{t:k'+1}(\mathcal{T}_\rho) N_{k',v}\rangle_H \left(P_{vv}^{2t-k-k'+2} - \frac{1}{n}\right)$$

$$\leq \sum_{k,k'=1}^{t} \eta_k \eta_{k'} \mathbf{E}|\langle \mathcal{T}_\rho^{1/2} \Pi_{t:k+1}(T_\rho) N_{k,v}, \mathcal{T}_\rho^{1/2} \Pi_{t:k'+1}(\mathcal{T}_\rho) N_{k',v}\rangle_H| \left|\left(P_{vv}^{2t-k-k'+2} - \frac{1}{n}\right)\right|$$

$$\leq \sum_{k,k'=1}^{t} \eta_k \eta_{k'} \mathbf{E}\left[\|\mathcal{T}_\rho^{1/2} \Pi_{t:k+1}(\mathcal{T}_\rho) N_{k,v}\|_H \|\mathcal{T}_\rho^{1/2} \Pi_{t:k'+1}(\mathcal{T}_\rho) N_{k',v}\|_H\right] \sigma_2^{2t-k-k'+2}$$

$$= \mathbf{E}\left[\left(\sum_{k=1}^{t} \eta_k \sigma_2^{t-k+1} \|\mathcal{T}_\rho^{1/2} \Pi_{t:k+1}(\mathcal{T}_\rho) N_{k,v}\|_H\right)^2\right],$$

where we used Jensen's inequality and the Cauchy-Schwarz inequality. $\qquad \square$

The following lemma presents the final bound for the **Population Covariance Error**. This result is established by utilising the series bound in Proposition 4 to split the error into well-mixed and poorly-mixed terms, i.e. for $k$ such that $t - k \gtrsim 1/(1 - \sigma_2)$ and $t - k \lesssim 1/(1 - \sigma_2)$. The well-mixed terms are controlled using that $\sigma_2^{t-k+1}$ is small. The poorly-mixed terms (there are $\sim 1/(1 - \sigma_2)$ of them) are controlled using both the concentration of the error terms $\{N_{k,w}\}_{k \geq 1, w \in V}$ as well as the contractive nature of the gradient updates, i.e. the operator norm of $\{\mathcal{T}_\rho^{1/2} \Pi_{t:k+1}(\mathcal{T}_\rho)\}$ in Lemma 7. The contractive terms arising from the gradient updates are decreasing in the step size: larger steps achieve a faster contraction. However, each term within the Network Error series is scaled by the step size $\{\eta_k\}_{k \geq 1}$, i.e. the Network Error takes the form $\sum_{k=1}^{t} \sigma_2^{t-k+1} \eta_k [\cdots]$ where $[\cdots]$ indicates the right most terms. To exploit this trade-off we introduce two free parameters $\alpha \in [0, 1/2]$ and $\gamma' \in [1, \gamma]$, which describe the degree to which the contraction is utilised. Specifically, $\alpha = 0$ and $\gamma' = \gamma$ is the large step regime and, $\alpha = 1/2$ and $\gamma' = 1$ is the small step regime.

**Lemma 8.** *Let Assumptions 1, 2, 3 hold with $r \geq 1/2$, $\eta_t = \eta t^{-\theta}$ for $t \in \mathbb{N}$ with $\eta \kappa^2 \leq 1$ and $\theta \in [0, 1)$. The following holds for any $v \in V$, $t/2 \geq \lceil \frac{(1+r) \log(t)}{1 - \sigma_2} \rceil =: t^\star$, $\alpha \in [0, 1/2]$ and $\gamma' \in [1, \gamma]$:*

$$
\mathbf{E}\left[ \left\| \sum_{k=1}^{t} \eta_k \sum_{w_{t:k} \in V^{t-k+1}} \left( P_{v w_{t:k}} - \frac{1}{n^{t-k+1}} \right) \mathcal{T}_\rho^{1/2} \Pi_{t:k+1}(\mathcal{T}_\rho) N_{k,w_k} \right\|_H^2 \right]
$$
$$
\leq \frac{\widetilde{a} \log^2(4n) \log^2(t^\star)}{m} \left( \eta^2 t^{-2r} \vee (m^{-1}(\eta t^\star)^{1+2\alpha}) \vee (\eta t^\star)^{\gamma'+2\alpha} \right),
$$

*where*
$$
\widetilde{a} = \frac{1152(R\kappa^{2r} + \sqrt{M})^2 (\kappa + \sqrt{2\sqrt{\nu} c_{\gamma'}})^2 (\|\mathcal{T}_\rho\| \vee 1)^2}{\|\mathcal{T}_\rho\| \wedge \|\mathcal{T}_\rho\|^{\gamma'}} \left[ 6 \left( \frac{\|\mathcal{T}_\rho^\alpha\| t^{-\alpha\theta}}{\alpha} \vee \frac{t^{-(\alpha+1/2)\theta} \|\mathcal{T}_\rho^\alpha\|}{1/2 + \alpha} \vee t^{-\theta} \|\mathcal{T}_\rho\| \right) \mathbb{1}_{\{\alpha \neq 0\}} + 10 \right]^2.
$$

*Proof.* Consider the bound of **Population Covariance Error** in Proposition 4. Let $\|\mathcal{T}_\rho\| \geq \lambda \geq 0$, $\widetilde{\lambda} \geq 0$ and for $c > 0$ introduce the cutoff $t^\star = \lceil \frac{c \log(t)}{1 - \sigma_2} \rceil$. For $k = 1, \ldots, t$ and $v \in V$ we have

$$
\|\mathcal{T}_\rho^{1/2} \Pi_{t:k+1}(\mathcal{T}_\rho) N_{k,v}\|_H \leq \|\mathcal{T}_\rho^{1/2} \Pi_{t:k+1}(\mathcal{T}_\rho) \mathcal{T}_{\rho,\lambda}^{1/2}\| \|\mathcal{T}_{\rho,\lambda}^{-1/2} N_{k,v}\|_H
$$
$$
\leq \|\mathcal{T}_\rho^{1/2} \Pi_{t:k+1}(\mathcal{T}_\rho) \mathcal{T}_{\rho,\lambda}^{1/2}\| \max_{k=1,\ldots,t} \left\{ \|\mathcal{T}_{\rho,\lambda}^{-1/2} N_{k,v}\|_H \right\},
$$

and similarly for $\widetilde{\lambda}$. Let us split the summation at $k \leq t - t^\star - 1$ and $k \geq t - t^\star$ using the bound above to obtain

$$
\left( \sum_{k=1}^{t} \sigma_2^{t-k+1} \eta_k \|\mathcal{T}_\rho^{1/2} \Pi_{t:k+1}(\mathcal{T}_\rho) N_{k,v}\|_H \right)^2
$$
$$
\leq 2 \underbrace{\left( \sum_{k=1}^{t-t^\star-1} \sigma_2^{t-k+1} \eta_k \|\mathcal{T}_\rho^{1/2} \Pi_{t:k+1}(\mathcal{T}_\rho) \mathcal{T}_{\rho,\lambda}^{1/2}\| \right)^2}_{\textbf{Well-Mixed Network Error}} \max_{k=1,\ldots,t} \left\{ \|\mathcal{T}_{\rho,\lambda}^{-1/2} N_{k,v}\|_H^2 \right\}
$$
$$
+ 2 \underbrace{\left( \sum_{k=t-t^\star}^{t} \sigma_2^{t-k+1} \eta_k \|\mathcal{T}_\rho^{1/2} \Pi_{t:k+1}(\mathcal{T}_\rho) \mathcal{T}_{\rho,\widetilde{\lambda}}^{1/2}\| \right)^2}_{\textbf{Poorly-Mixed Network Error}} \max_{k=1,\ldots,t} \left\{ \|\mathcal{T}_{\rho,\widetilde{\lambda}}^{-1/2} N_{k,v}\|_H^2 \right\}.
$$

The **Well-Mixed Network Error** is controlled through $\sigma_2^{t-k+1}$ being small for $k \leq t - t^\star$. From $\|\Pi_{t:k+1}(\mathcal{T}_\rho)\| \leq 1$ and $\lambda \leq \|\mathcal{T}_\rho\|$ we have $\|\mathcal{T}_\rho^{1/2} \Pi_{t:k+1}(\mathcal{T}_\rho) \mathcal{T}_{\rho,\lambda}^{1/2}\|_H \leq 2\|\mathcal{T}_\rho\|$, and from $1/\log(1/\sigma_2) \leq 1/(1 - \sigma_2)$ we have $t^\star \geq \frac{c \log(t)}{-\log(\sigma_2)}$. These two facts allow the **Well-Mixed Network Error** to be bounded as follows:

$$
\textbf{Well-Mixed Network Error} \leq 2\|\mathcal{T}_\rho\| \eta \sum_{k=1}^{t-t^\star} \sigma_2^{t-k+1} k^{-\theta} \leq 2\eta\|\mathcal{T}_\rho\| \sum_{k=1}^{t-t^\star} \sigma_2^{\frac{c \log(t)}{-\log(\sigma_2)}} \leq 2\eta\|\mathcal{T}_\rho\| t^{1-c}.
$$

For the **Poorly-Mixed Network Error** let us consider the two cases $\alpha \in (0, 1/2]$ and $\alpha = 0$ separately. Consider $\alpha \in (0, 1/2]$ first. Using Lemma 7[4] we have, for $t - 1 \geq k \geq 1$,

$$\|\mathcal{T}_\rho^{1/2} \Pi_{t:k+1}(\mathcal{T}_\rho) \mathcal{T}_{\rho,\widetilde{\lambda}}^{1/2}\| \leq \|\mathcal{T}_\rho \Pi_{t:k+1}(\mathcal{T}_\rho)\| + \sqrt{\widetilde{\lambda}} \|\mathcal{T}_\rho^{1/2} \Pi_{t:k+1}(\mathcal{T}_\rho)\|$$

$$\leq \|\mathcal{T}_\rho^\alpha\| \|\mathcal{T}_\rho^{1-\alpha} \Pi_{t:k+1}(\mathcal{T}_\rho)\| + \sqrt{\widetilde{\lambda}} \|\mathcal{T}_\rho^\alpha\| \|\mathcal{T}_\rho^{1/2-\alpha} \Pi_{t:k+1}(\mathcal{T}_\rho)\|$$

$$\leq \|\mathcal{T}_\rho^\alpha\| \left( \frac{1-\alpha}{e \sum_{j=k+1}^t \eta_j} \right)^{1-\alpha} + \sqrt{\widetilde{\lambda}} \|\mathcal{T}_\rho^\alpha\| \left( \frac{1/2-\alpha}{e \sum_{j=k+1}^t \eta_j} \right)^{1/2-\alpha}.$$

When plugging the above into the **Poorly-Mixed Network Error**, summations of the form $\sum_{k=t-t^\star}^{t-1} \frac{\eta_k}{(\sum_{j=k+1}^t \eta_j)^\beta}$ appear for $\beta = 1 - \alpha$ and $\beta = 1/2 - \alpha$. To bound these consider the following for $\beta \in [0, 1)$ and $t \geq 2t^\star$:

$$\sum_{k=t-t^\star}^{t-1} \frac{\eta_k}{\left( \sum_{j=k+1}^t \eta_j \right)^\beta} = \eta^{1-\beta} \sum_{k=t-t^\star}^{t-1} \frac{k^{-\theta}}{\left( \sum_{j=k+1}^t j^{-\theta} \right)^\beta}$$

$$\leq \eta^{1-\beta} t^{\theta\beta} \sum_{k=t-t^\star}^{t-1} \frac{k^{-\theta}}{\left( t - k \right)^\beta}$$

$$\leq \frac{\eta^{1-\beta} t^{\theta\beta}}{(t - t^\star)^\theta} \sum_{k=t-t^\star}^{t-1} \frac{1}{\left( t - k \right)^\beta}$$

$$= \frac{\eta^{1-\beta} t^{\theta\beta}}{(t - t^\star)^\theta} \sum_{k=1}^{t^\star} \frac{1}{k^\beta}$$

$$\leq 2\eta^{1-\beta} t^{\theta(\beta-1)} \frac{(t^\star)^{1-\beta}}{1 - \beta},$$

where the last inequality follows from an integral bound as well as using that $\frac{t^{\theta\beta}}{(t-t^\star)^\theta} = \frac{t^{\theta(\beta-1)}}{(1-\frac{t^\star}{t})^\theta} \leq 2t^{\theta(\beta-1)}$ from $t \geq 2t^\star$. Splitting the summation at $k = t$, plugging the above two bounds into the **Poorly-Mixed Network Error** term and using $(\eta t^\star)^\alpha \geq \eta$ from $\eta \leq \kappa^{-2} \leq 1$ yields a bound for $\alpha \in (0, 1/2]$:

**Poorly-Mixed Network Error**

$$\leq \frac{2\|\mathcal{T}_\rho^\alpha\| t^{-\alpha\theta}}{\alpha} (\eta t^\star)^\alpha + \frac{2t^{-(\alpha+1/2)\theta} \|\mathcal{T}_\rho^\alpha\|}{1/2 + \alpha} \sqrt{\widetilde{\lambda}} (\eta t^\star)^{1/2+\alpha} + \sqrt{2} \eta t^{-\theta} \|\mathcal{T}_\rho\|$$

$$\leq 6 \left( \frac{\|\mathcal{T}_\rho^\alpha\| t^{-\alpha\theta}}{\alpha} \vee \frac{t^{-(\alpha+1/2)\theta} \|\mathcal{T}_\rho^\alpha\|}{1/2 + \alpha} \vee t^{-\theta} \|\mathcal{T}_\rho\| \right) \left( (\eta t^\star)^\alpha \vee \sqrt{\widetilde{\lambda}} (\eta t^\star)^{1/2+\alpha} \right).$$

Now consider the case $\alpha = 0$. The summation for $\beta = 1$ in this case is bounded following the previous steps

$$\sum_{k=t-t^\star}^{t-1} \frac{\eta_k}{\left( \sum_{j=k+1}^t \eta_j \right)} \leq \frac{t^\theta}{(t - t^\star)^\theta} \sum_{k=t-t^\star}^{t-1} \frac{1}{(t - k)} \leq 2^{1+\theta} \log(t^\star),$$

leading to the **Poorly-Mixed Network Error** bounded as for $\alpha = 0$ from $\eta \|\mathcal{T}_\rho\| \leq 1$:

**Poorly-Mixed Network Error** $\leq 2^{1+\theta} \log(t^\star) + 4t^{-\theta/2} \sqrt{\widetilde{\lambda}} (\eta t^\star)^{1/2} + \sqrt{2} \eta t^{-\theta} \|\mathcal{T}_\rho\|$

$$\leq 10 \log(t^\star)(1 \vee (\sqrt{\widetilde{\lambda}} (\eta t^\star)^{1/2})).$$

Combining the two bounds for $\alpha = 0$ and $\alpha \in (0, 1/2]$ gives

**Poorly-Mixed Network Error**

$$\leq \log(t^\star)\left[6\left(\frac{\|\mathcal{T}_\rho^\alpha\|t^{-\alpha\theta}}{\alpha} \vee \frac{t^{-(\alpha+1/2)\theta}\|\mathcal{T}_\rho^\alpha\|}{1/2+\alpha} \vee t^{-\theta}\|\mathcal{T}_\rho\|\right)\mathbb{1}_{\{\alpha \neq 0\}} + 10\right]\left((\eta t^\star)^\alpha \vee \sqrt{\widetilde{\lambda}}(\eta t^\star)^{1/2+\alpha}\right).$$

We now consider the terms $\max_{k=1,\ldots,t}\{\|\mathcal{T}_{\rho,\lambda}^{-1/2}N_{k,v}\|_H^2\}$ for both $\lambda$ and $\widetilde{\lambda}$. We use the high-probability bounds of Lemma 6 to uniformly control $\|\mathcal{T}_{\rho,\lambda}^{-1/2}N_{k,v}\|_H^2$ for all $k = 1, \ldots, t$ and $v \in V$. For $w \in V$, let $\delta_w = \frac{\delta}{n}$. With probability at least $1 - \delta_w$ the following holds for all $k = 1, \ldots, t$ and $\gamma' \in [1, \gamma]$:

$$\|\mathcal{T}_{\rho,\lambda}^{-1/2}N_{k,w}\|_H^2 \leq 16(R\kappa^{2r} + \sqrt{M})^2\left(\frac{\kappa}{m\sqrt{\lambda}} + \frac{\sqrt{2\sqrt{\nu}c_{\gamma'}}}{\sqrt{m\lambda^{\gamma'}}}\right)^2\log^2\frac{4n}{\delta}.$$

We note that if the capacity assumption holds for $\gamma$, then it also holds for all $\gamma' \in [1, \gamma]$. Applying a union bound, we get that the above holds with probability at least $1 - \sum_{v \in V}\delta_v = 1 - \delta$ for all $w \in V$ and $k = 1, \ldots, t$. Using Lemma 4, the expectation of the maximum can be bounded for any $v \in V$ and $\gamma' \in [1, \gamma]$ as follows:

$$\mathbf{E}\left[\max_{k=1,\ldots,t}\{\|\mathcal{T}_{\rho,\lambda}^{-1/2}N_{k,v}\|_H^2\}\right]$$

$$\leq 16(R\kappa^{2r} + \sqrt{M})^2\left(\frac{\kappa}{m\sqrt{\lambda}} + \frac{\sqrt{2\sqrt{\nu}c_{\gamma'}}}{\sqrt{m\lambda^{\gamma'}}}\right)^2\int_0^1\log^2\frac{4n}{\delta}d\delta$$

$$\leq 96(R\kappa^{2r} + \sqrt{M})^2\left(\frac{\kappa}{m\sqrt{\lambda}} + \frac{\sqrt{2\sqrt{\nu}c_{\gamma'}}}{\sqrt{m\lambda^{\gamma'}}}\right)^2\log^2 4n,$$

where we used $\int_0^1\log^2\frac{4n}{\delta}d\delta \leq 6\log^2 4n$.

Bringing together the bounds for the **Poorly-** and **Well-Mixed Network Error** with the above bound for the quantity $\mathbf{E}\left[\max_{k=1,\ldots,t}\{\|\mathcal{T}_{\rho,\lambda}^{-1/2}N_{k,v}\|_H^2\}\right]$ yields

$$\mathbf{E}\left[\left(\sum_{k=1}^t\sigma_2^{t-k+1}\eta_k\|\mathcal{T}_\rho^{1/2}\Pi_{t:k+1}(\mathcal{T}_\rho)N_{k,v}\|_H\right)^2\right]$$

$$\leq 96\log^2(4n)\log^2(t^\star)(R\kappa^{2r} + \sqrt{M})^2$$

$$\times \left(8\|\mathcal{T}_\rho\|^2\left(\frac{\kappa}{m\sqrt{\lambda}} + \frac{\sqrt{2\sqrt{\nu}c_\gamma}}{\sqrt{m\lambda^\gamma}}\right)^2\eta^2 t^{2(1-c)}\right.$$

$$+ 2\left[6\left(\frac{\|\mathcal{T}_\rho^\alpha\|t^{-\alpha\theta}}{\alpha} \vee \frac{t^{-(\alpha+1/2)\theta}\|\mathcal{T}_\rho^\alpha\|}{1/2+\alpha} \vee t^{-\theta}\|\mathcal{T}_\rho\|\right)\mathbb{1}_{\{\alpha \neq 0\}} + 10\right]^2\left(\frac{\kappa}{m\sqrt{\widetilde{\lambda}}} + \frac{\sqrt{2\sqrt{\nu}c_{\gamma'}}}{\sqrt{m\widetilde{\lambda}^{\gamma'}}}\right)^2$$

$$\times \left.\left((\eta t^\star)^{2\alpha} \vee \widetilde{\lambda}(\eta t^\star)^{1+2\alpha}\right)\right).$$

Let $\lambda = \|\mathcal{T}_\rho\|$ and $\widetilde{\lambda} = \frac{\|\mathcal{T}_\rho\|}{\eta t^\star}$. The bound

$$\frac{1}{m\sqrt{\widetilde{\lambda}}} + \frac{1}{\sqrt{m\widetilde{\lambda}^{\gamma'}}} \leq \frac{2}{\sqrt{m}}\left(\frac{1}{\sqrt{m\|\mathcal{T}_\rho\|(\eta t^\star)^{-1}}} \vee \frac{1}{\|\mathcal{T}_\rho\|^{\gamma'/2}(\eta t^\star)^{-\gamma'/2}}\right)$$

$$\leq \frac{2}{\sqrt{m(\|\mathcal{T}_\rho\| \wedge \|\mathcal{T}_\rho\|^{\gamma'})}}\left(\sqrt{\eta t^\star/m} \vee (\eta t^\star)^{\gamma'/2}\right)$$

allows the expected squared series to be bounded as follows:

$$\mathbf{E}\left[\left(\sum_{k=1}^t\sigma_2^{t-k+1}\eta_k\|\mathcal{T}_\rho^{1/2}\Pi_{t:k+1}(\mathcal{T}_\rho)N_{k,v}\|_H\right)^2\right]$$

$$\leq \frac{\widetilde{a}\log^2(4n)\log^2(t^\star)}{m}\left((\eta t^{1-c})^2 \vee (m^{-1}(\eta t^\star)^{1+2\alpha}) \vee (\eta t^\star)^{\gamma'+2\alpha}\right)$$

where
$$\widetilde{a} = \frac{1152(R\kappa^{2r}+\sqrt{M})^2(\kappa+\sqrt{2\sqrt{\nu}c_{\gamma'}})^2(\|\mathcal{T}_\rho\|\vee 1)^2}{\|\mathcal{T}_\rho\|\wedge\|\mathcal{T}_\rho\|^{\gamma'}}\left[6\left(\frac{\|\mathcal{T}_\rho^\alpha\|t^{-\alpha\theta}}{\alpha}\vee\frac{t^{-(\alpha+1/2)\theta}\|\mathcal{T}_\rho^\alpha\|}{1/2+\alpha}\vee t^{-\theta}\|\mathcal{T}_\rho\|\right)\mathbb{1}_{\{\alpha\neq 0\}}+10\right]^2.$$

The choice $c = 1 + r$ yields the final result. $\qquad\square$

### C.3.2 Analysis of Residual Empirical Covariance Error

In this section we develop a bound for the **Residual Empirical Covariance Error** term in (9). The final result is presented in Lemma 9.

The following proposition writes the **Residual Empirical Covariance Error** in terms of a series of quantities that will be later controlled.

**Proposition 5.** *Let $t \geq k + 1$. For any $w_{t:k+1} \in V^{t-k}$ we have*

$$\Pi_{t:k+1}(\mathcal{T}_{\mathbf{x}_{w_{t:k+1}}}) = \Pi_{t:k+1}(\mathcal{T}_\rho) + \sum_{j=k+1}^{t} \eta_j \Pi_{t:j+1}(\mathcal{T}_\rho)(\mathcal{T}_\rho - \mathcal{T}_{\mathbf{x}_{w_j}})\Pi_{j-1:k+1}(\mathcal{T}_{\mathbf{x}_{w_{j-1:k+1}}}).$$

*Proof.* Adding and subtracting $(I - \eta_t\mathcal{T}_\rho)\Pi_{t-1:k+1}(\mathcal{T}_{\mathbf{x}_{w_{t-1:k+1}}})$ and unravelling yields the following:

$$\begin{aligned}
&\Pi_{t:k+1}(\mathcal{T}_{\mathbf{x}_{w_{t:k+1}}}) - \Pi_{t:k+1}(\mathcal{T}_\rho)\\
&= (I - \eta_t\mathcal{T}_{\mathbf{x}_{w_t}})\Pi_{t-1:k+1}(\mathcal{T}_{\mathbf{x}_{w_{t-1:k+1}}}) - (I - \eta_t\mathcal{T}_\rho)\Pi_{t-1:k+1}(\mathcal{T}_\rho)\\
&= (I - \eta_t\mathcal{T}_{\mathbf{x}_{w_t}})\Pi_{t-1:k+1}(\mathcal{T}_{\mathbf{x}_{w_{t-1:k+1}}}) - (I - \eta_t\mathcal{T}_\rho)\Pi_{t-1:k+1}(\mathcal{T}_{\mathbf{x}_{w_{t-1:k+1}}})\\
&\quad + (I - \eta_t\mathcal{T}_\rho)\Pi_{t-1:k+1}(\mathcal{T}_{\mathbf{x}_{w_{t-1:k+1}}}) - (I - \eta_t\mathcal{T}_\rho)\Pi_{t-1:k+1}(\mathcal{T}_\rho)\\
&= \eta_t(\mathcal{T}_\rho - \mathcal{T}_{\mathbf{x}_{w_t}})\Pi_{t-1:k+1}(\mathcal{T}_{\mathbf{x}_{w_{t-1:k+1}}}) + (I - \eta_t\mathcal{T}_\rho)\left[\Pi_{t-1:k+1}(\mathcal{T}_{\mathbf{x}_{w_{t-1:k+1}}}) - \Pi_{t-1:k+1}(\mathcal{T}_\rho)\right]\\
&= \sum_{j=k+1}^{t} \eta_j \Pi_{t:j+1}(\mathcal{T}_\rho)(\mathcal{T}_\rho - \mathcal{T}_{\mathbf{x}_{w_j}})\Pi_{j-1:k+1}(\mathcal{T}_{\mathbf{x}_{w_{j-1:k+1}}}).
\end{aligned}$$

$\qquad\square$

Applying Proposition 5 to the **Residual Empirical Covariance Error** term, using the triangle equality, yields

$$\left\|\sum_{k=1}^{t}\eta_k\sum_{w_{t:k}\in V^{t-k+1}}\Delta(w_{t:k})\mathcal{T}_\rho^{1/2}\big(\Pi_{t:k+1}(\mathcal{T}_{\mathbf{x}_{w_{t:k+1}}}) - \Pi_{t:k+1}(\mathcal{T}_\rho)\big)N_{k,w_k}\right\|_H$$

$$\leq \sum_{k=1}^{t-1}\eta_k\sum_{w_{t:k}\in V^{t-k+1}}|\Delta(w_{t:k})|\sum_{j=k+1}^{t}\eta_j$$

$$\times\|\mathcal{T}_\rho^{1/2}\Pi_{t:j+1}(\mathcal{T}_\rho)(\mathcal{T}_\rho - \mathcal{T}_{\mathbf{x}_{w_j}})\Pi_{j-1:k+1}(\mathcal{T}_{\mathbf{x}_{w_{j-1:k+1}}})N_{k,w_k}\|_H, \tag{12}$$

where the quantity is zero in the case $k = t$. For $j \in \{2, \ldots, t-1\}$ the above includes the quantity $\Pi_{t:j+1}(\mathcal{T}_\rho)$. This can be interpreted in a similar manner to the filter function associated for gradient descent, see for instance [26, Example 2]. In this context it is used to control the growth of the above error term, which is absent in the case $j = t$. This yields the following proposition.

**Proposition 6.** *Let Assumptions 1, 2, 3 hold with $r \geq 1/2$ and $\eta_t = \eta t^{-\theta}$ for $t \in \mathbb{N}$ with $\eta\kappa^2 \leq 1$, $\theta \in (0,1)$. Fix $\lambda, \widetilde{\lambda} > 0$ and $\delta \in (0,1)$. With probability at least $1 - \delta$ the following hold: for any $t - 1 \geq j \geq k + 1$ and path $w_{t:k} \in V^{t-k+1}$ we have*

$$\|\mathcal{T}_\rho^{1/2}\Pi_{t:j+1}(\mathcal{T}_\rho)(\mathcal{T}_\rho - \mathcal{T}_{\mathbf{x}_{w_j}})\Pi_{j-1:k+1}(\mathcal{T}_{\mathbf{x}_{w_{j-1:k+1}}})N_{k,w_k}\|_H$$

$$\leq 2\kappa\|\mathcal{T}_{\rho,\widetilde{\lambda}}^{1/2}\|\left(\frac{1}{\sum_{i=j+1}^{t}\eta_i} + \left(\frac{\lambda}{\sum_{i=j+1}^{t}\eta_i}\right)^{1/2}\right)\left(\frac{2\kappa}{m\sqrt{\lambda}} + \frac{\sqrt{c_\gamma}}{\sqrt{m\lambda^\gamma}}\right)\log\left(\frac{4n}{\delta}\right)$$

$$\times \max_{w\in V}\left\{\|\mathcal{T}_{\rho,\widetilde{\lambda}}^{-1/2}N_{k,w}\|_H\right\}, \tag{13}$$

*for any* $t - 1 \geq k \geq 1$ *and nodes* $w_t, w_k \in V$

$$\|\mathcal{T}_\rho^{1/2}(\mathcal{T}_\rho - \mathcal{T}_{\mathbf{x}_{w_t}})N_{k,w_k}\|_H$$
$$\leq 2\kappa \|\mathcal{T}_\rho^{1/2}\mathcal{T}_{\rho,\lambda}^{1/2}\|\|\mathcal{T}_{\rho,\widetilde{\lambda}}^{1/2}\|\left(\frac{2\kappa}{m\sqrt{\lambda}} + \frac{\sqrt{c_\gamma}}{\sqrt{m\lambda^\gamma}}\right) \log \frac{4n}{\delta} \max_{w\in V}\left\{\|\mathcal{T}_{\rho,\widetilde{\lambda}}^{-1/2}N_{k,w}\|_H\right\}. \qquad (14)$$

*Proof.* Fix $t - 1 \geq j \geq k + 1$ and $w_{t:k} \in V^{t-k+1}$. Begin by proving (13). Expanding the norm,

$$\|\mathcal{T}_\rho^{1/2}\Pi_{t:j+1}(\mathcal{T}_\rho)(\mathcal{T}_\rho - \mathcal{T}_{\mathbf{x}_{w_j}})\Pi_{j-1:k+1}(\mathcal{T}_{\mathbf{x}_{w_{j-1:k+1}}})N_{k,w_k}\|_H$$
$$= \|\mathcal{T}_\rho^{1/2}\Pi_{t:j+1}(\mathcal{T}_\rho)\mathcal{T}_{\rho,\lambda}^{1/2}\mathcal{T}_{\rho,\lambda}^{-1/2}(\mathcal{T}_\rho - \mathcal{T}_{\mathbf{x}_{w_j}})\Pi_{j-1:k+1}(\mathcal{T}_{\mathbf{x}_{w_{j-1:k+1}}})\mathcal{T}_{\rho,\widetilde{\lambda}}^{1/2}\mathcal{T}_{\rho,\widetilde{\lambda}}^{-1/2}N_{k,w_k}\|_H$$
$$\leq \|\mathcal{T}_\rho^{1/2}\Pi_{t:j+1}(\mathcal{T}_\rho)\mathcal{T}_{\rho,\lambda}^{1/2}\|\|\mathcal{T}_{\rho,\lambda}^{-1/2}(\mathcal{T}_\rho - \mathcal{T}_{\mathbf{x}_{w_j}})\|\|\Pi_{j-1:k+1}(\mathcal{T}_{\mathbf{x}_{w_{j-1:k+1}}})\|\|\mathcal{T}_{\rho,\widetilde{\lambda}}^{1/2}\|\|\mathcal{T}_{\rho,\widetilde{\lambda}}^{-1/2}N_{k,w_k}\|_H$$
$$\leq \|\mathcal{T}_\rho^{1/2}\Pi_{t:j+1}(\mathcal{T}_\rho)\mathcal{T}_{\rho,\lambda}^{1/2}\|\|\mathcal{T}_{\rho,\lambda}^{-1/2}(\mathcal{T}_\rho - \mathcal{T}_{\mathbf{x}_{w_j}})\|\|\mathcal{T}_{\rho,\widetilde{\lambda}}^{1/2}\|\|\mathcal{T}_{\rho,\widetilde{\lambda}}^{-1/2}N_{k,w_k}\|_H$$
$$\leq \|\mathcal{T}_\rho^{1/2}\Pi_{t:j+1}(\mathcal{T}_\rho)\mathcal{T}_{\rho,\lambda}^{1/2}\|\|\mathcal{T}_{\rho,\lambda}^{-1/2}(\mathcal{T}_\rho - \mathcal{T}_{\mathbf{x}_{w_j}})\|\|\mathcal{T}_{\rho,\widetilde{\lambda}}^{1/2}\| \max_{w\in V}\left\{\|\mathcal{T}_{\rho,\widetilde{\lambda}}^{-1/2}N_{k,w}\|_H\right\},$$

where we used, from $\eta\kappa^2 \leq 1$ and $\eta\|\mathcal{T}_{\mathbf{x}_v}\| \leq 1$ for any $v \in V$, that $\|\Pi_{j-1:k+1}(\mathcal{T}_{\mathbf{x}_{w_{j-1:k+1}}})\| \leq 1$ for $j \geq k + 2$. The first operator norm is bounded as follows by using techniques similar to those used to prove Lemma 7:

$$\|\mathcal{T}_\rho^{1/2}\Pi_{t:j+1}(\mathcal{T}_\rho)\mathcal{T}_{\rho,\lambda}^{1/2}\| \leq \left(\frac{1}{e\sum_{i=j+1}^t \eta_i} + \left(\frac{\lambda}{2e\sum_{i=j+1}^t \eta_i}\right)^{1/2}\right)$$
$$\leq \left(\frac{1}{\sum_{i=j+1}^t \eta_i} + \left(\frac{\lambda}{\sum_{i=j+1}^t \eta_i}\right)^{1/2}\right). \qquad (15)$$

We proceed to construct a high-probability bound for the quantity $\|(\mathcal{T}_\rho + \lambda I)^{-1/2}(\mathcal{T}_\rho - \mathcal{T}_{\mathbf{x}_{w_j}})\|$, for any $w_j \in V$. For $v \in V$, let $\delta_v = \frac{\delta}{n}$ and apply (11) from Lemma 6 to obtain the following[5] with probability at least $1 - \delta_v$:

$$\|(\mathcal{T}_\rho + \lambda I)^{-1/2}(\mathcal{T}_\rho - \mathcal{T}_{\mathbf{x}_v})\| \leq \|(\mathcal{T}_\rho + \lambda I)^{-1/2}(\mathcal{T}_\rho - \mathcal{T}_{\mathbf{x}_v})\|_{HS} \leq 2\kappa\left(\frac{2\kappa}{m\sqrt{\lambda}} + \frac{\sqrt{c_\gamma}}{\sqrt{m\lambda^\gamma}}\right)\log\frac{4n}{\delta}.$$

Applying a union bound yields the following with probability at least $1 - \sum_{v\in V}\delta_v = 1 - \delta$:

$$\|(\mathcal{T}_\rho + \lambda I)^{-1/2}(\mathcal{T}_\rho - \mathcal{T}_{\mathbf{x}_v})\| \leq 2\kappa\left(\frac{2\kappa}{m\sqrt{\lambda}} + \frac{\sqrt{c_\gamma}}{\sqrt{m\lambda^\gamma}}\right)\log\frac{4n}{\delta} \quad \forall v \in V. \qquad (16)$$

The result (13) then comes from plugging (15) and (16) into the expanded quantity at the start of the proof.

To prove (14), fix $t - 1 \geq k \geq 1$ and $w_t, w_k \in V$. Expanding the norm we get

$$\|\mathcal{T}_\rho^{1/2}(\mathcal{T}_\rho - \mathcal{T}_{\mathbf{x}_{w_t}})N_{k,w_k}\|_H = \|\mathcal{T}_\rho^{1/2}\mathcal{T}_{\rho,\lambda}^{1/2}\mathcal{T}_{\rho,\lambda}^{-1/2}(\mathcal{T}_\rho - \mathcal{T}_{\mathbf{x}_{w_t}})\mathcal{T}_{\rho,\widetilde{\lambda}}^{1/2}\mathcal{T}_{\rho,\widetilde{\lambda}}^{-1/2}N_{k,w_k}\|_H$$
$$\leq \|\mathcal{T}_\rho^{1/2}\mathcal{T}_{\rho,\lambda}^{1/2}\|\|\mathcal{T}_{\rho,\lambda}^{-1/2}(\mathcal{T}_\rho - \mathcal{T}_{\mathbf{x}_{w_t}})\|\|\mathcal{T}_{\rho,\widetilde{\lambda}}^{1/2}\|\|\mathcal{T}_{\rho,\widetilde{\lambda}}^{-1/2}N_{k,w_k}\|_H$$
$$\leq \|\mathcal{T}_\rho^{1/2}\mathcal{T}_{\rho,\lambda}^{1/2}\|\|\mathcal{T}_{\rho,\lambda}^{-1/2}(\mathcal{T}_\rho - \mathcal{T}_{\mathbf{x}_{w_t}})\|\|\mathcal{T}_{\rho,\widetilde{\lambda}}^{1/2}\| \max_{w\in V}\left\{\|\mathcal{T}_{\rho,\widetilde{\lambda}}^{-1/2}N_{k,w}\|_H\right\}.$$

The result follows by using (16) to bound $\|\mathcal{T}_{\rho,\lambda}^{-1/2}(\mathcal{T}_\rho - \mathcal{T}_{\mathbf{x}_{w_t}})\|$. $\qquad\qquad\square$

The following proposition utilise the previous proposition to bound the summation (12).

**Proposition 7.** *Let the assumptions of Proposition 6 hold. For any $v \in V$, with probability at least $1 - \delta$ we have*

$$\textbf{\textit{Resid. Emp. Cov. Error}} \leq 8\kappa \left( \frac{2\kappa}{m\sqrt{\lambda}} + \frac{\sqrt{c_\gamma}}{\sqrt{m\lambda^\gamma}} \right) \log \frac{4n}{\delta} \left[ \boldsymbol{B}_1 + \boldsymbol{B}_2 \right],$$

*where*

$$\boldsymbol{B}_1 = \|\mathcal{T}_\rho^{1/2} \mathcal{T}_{\rho,\lambda}^{1/2}\| \|\mathcal{T}_{\rho,\widetilde{\lambda}}^{1/2}\| \eta_t \sum_{k=1}^{t-1} \eta_k \max_{w \in V} \left\{ \|\mathcal{T}_{\rho,\widetilde{\lambda}}^{-1/2} N_{k,w}\|_H \right\},$$

$$\boldsymbol{B}_2 = \|\mathcal{T}_{\rho,\widetilde{\lambda}}^{1/2}\| \sum_{k=1}^{t-2} \eta_k \sum_{j=k+1}^{t-1} \eta_j \left( \frac{1}{\sum_{i=j+1}^{t} \eta_i} + \left( \frac{\lambda}{\sum_{i=j+1}^{t} \eta_i} \right)^{1/2} \right) \max_{w \in V} \left\{ \|\mathcal{T}_{\rho,\widetilde{\lambda}}^{-1/2} N_{k,w}\|_H \right\}.$$

*Proof.* Splitting the sum in (12) at $j = t$ and otherwise, directly applying (13) and (14) from Proposition 6 allows **Resid. Emp. Cov. Error** to be bounded as follows:

**Resid. Emp. Cov. Error**

$$\leq \eta_t \sum_{k=1}^{t-1} \eta_k \sum_{w_{t:k} \in V^{t-k+1}} |\Delta(w_{t:k})| \|\mathcal{T}_\rho^{1/2} (\mathcal{T}_\rho - \mathcal{T}_{\mathbf{x}_{w_t}}) \Pi_{t-1:k+1} (\mathcal{T}_{\mathbf{x}_{w_{t-1:k+1}}}) N_{k,w_k}\|_H$$

$$+ \sum_{k=1}^{t-2} \eta_k \sum_{w_{t:k} \in V^{t-k+1}} |\Delta(w_{t:k})| \sum_{j=k+1}^{t-1} \eta_j \|\mathcal{T}_\rho^{1/2} \Pi_{t:j+1}(\mathcal{T}_\rho)(\mathcal{T}_\rho - \mathcal{T}_{\mathbf{x}_{w_j}}) \Pi_{j-1:k+1}(\mathcal{T}_{\mathbf{x}_{w_{j-1:k+1}}}) N_{k,w_k}\|_H$$

$$\leq 2\kappa \left( \frac{2\kappa}{m\sqrt{\lambda}} + \frac{\sqrt{c_\gamma}}{\sqrt{m\lambda^\gamma}} \right) \log \frac{4n}{\delta}$$

$$\times \Bigg[ \underbrace{\|\mathcal{T}_\rho^{1/2} \mathcal{T}_{\rho,\lambda}^{1/2}\| \|\mathcal{T}_{\rho,\widetilde{\lambda}}^{1/2}\| \eta_t \sum_{k=1}^{t-1} \eta_k \max_{w \in V} \left\{ \|\mathcal{T}_{\rho,\widetilde{\lambda}}^{-1/2} N_{k,w}\|_H \right\} \sum_{w_{t:k} \in V^{t-k+1}} |\Delta(w_{t:k})|}_{\boldsymbol{B}_1}$$

$$+ \underbrace{\|\mathcal{T}_{\rho,\widetilde{\lambda}}^{1/2}\| \sum_{k=1}^{t-2} \eta_k \sum_{j=k+1}^{t-1} \eta_j \left( \frac{1}{\sum_{i=j+1}^{t} \eta_i} + \left( \frac{\lambda}{\sum_{i=j+1}^{t} \eta_i} \right)^{1/2} \right) \max_{w \in V} \left\{ \|\mathcal{T}_{\rho,\widetilde{\lambda}}^{-1/2} N_{k,w}\|_H \right\}}_{\boldsymbol{B}_2}$$

$$\times \sum_{w_{t:k} \in V^{t-k+1}} |\Delta(w_{t:k})| \Bigg].$$

The result is then arrived at by applying the following bound for the summation $\sum_{w_{t:k} \in V^{t-k+1}} |\Delta(w_{t:k})|$ for each $k \leq t$:

$$\sum_{w_{t:k} \in V^{t-k+1}} |\Delta(w_{t:k})| = \sum_{w_{t:k} \in V^{t-k+1}} \left| P_{vw_{t:k}} - \frac{1}{n^{t-k+1}} \right|$$

$$= \sum_{\substack{w_{t:k} \in V^{t-k+1} \\ P_{vw_{t:k}} \geq n^{-(t-k+1)}}} \left( P_{vw_{t:k}} - \frac{1}{n^{t-k+1}} \right) - \sum_{\substack{w_{t:k} \in V^{t-k+1} \\ P_{vw_{t:k}} < n^{-(t-k+1)}}} \left( P_{vw_{t:k}} - \frac{1}{n^{t-k+1}} \right) \leq 4.$$

$\square$

Given Proposition 7 we can now plug in a high-probability bound for $\max_{w \in V} \left\{ \|\mathcal{T}_{\rho,\widetilde{\lambda}}^{-1/2} N_{k,w}\|_H \right\}$ and bound the resulting summations. This is summarised in the following lemma.

**Lemma 9.** *Let the assumptions of Proposition 6 hold with $0 \leq \theta \leq 3/4$, $0 \leq \lambda \leq \|\mathcal{T}_\rho\|$ and $0 \leq \widetilde{\lambda} \leq \|\mathcal{T}_\rho\|$. Given $\delta \in (0,1)$, the following holds with probability at least $1 - \delta$:*

**Resid. Emp. Cov. Error**

$$\leq \widetilde{b}_1 \frac{\log^2 \frac{8n}{\delta} \log(t)}{m\sqrt{((m\lambda) \wedge \lambda^\gamma)((m\widetilde{\lambda}) \wedge \widetilde{\lambda}^\gamma)}} \left( 1 \vee (\eta t^{1-\theta}) \vee \sqrt{\lambda}(\eta t^{1-\theta})^{3/2} \vee (t^{-1}(\eta t^{1-\theta})^2) \right),$$

*where* $\widetilde{b}_1 = \frac{128\kappa(R\kappa^{2r}+\sqrt{M})(2\kappa+\sqrt{2\sqrt{\nu}c_\gamma})^2\|\mathcal{T}_\rho\|^{1/2}(4+\|\mathcal{T}_\rho\|)}{(1-\theta)}$.

*Proof.* Consider Proposition 7 with $\frac{\delta}{2}$, so the following holds with probability at least $1 - \frac{\delta}{2}$

$$\textbf{Resid. Emp. Cov. Error} \leq 8\kappa\left(\frac{2\kappa}{m\sqrt{\lambda}} + \frac{\sqrt{c_\gamma}}{\sqrt{m\lambda^\gamma}}\right)\log\frac{8n}{\delta}(\mathbf{B}_1 + \mathbf{B}_2)$$

$$\leq \frac{8\kappa(2\kappa + \sqrt{2\sqrt{\nu}c_\gamma})}{\sqrt{(m\lambda)\wedge\lambda^\gamma}}\frac{\log\frac{8n}{\delta}}{\sqrt{m}}(\mathbf{B}_1 + \mathbf{B}_2),$$

where we used that $\nu \geq 1$. Proceed to bound both $\mathbf{B}_1$ and $\mathbf{B}_2$. Start by constructing a high-probability bound for the term $\max_{w\in V}\left\{\|\mathcal{T}_{\rho,\widetilde{\lambda}}^{-1/2}N_{k,w}\|_H\right\}$ $k = 1,\ldots,t$. For $v \in V$, let $\delta'_v = \frac{\delta}{2n}$. Lemma 6 states with probability at least $1 - \delta'_v$ the following holds for any $k \in \mathbb{N}$:

$$\|\mathcal{T}_{\rho,\widetilde{\lambda}}^{-1/2}N_{k,v}\| \leq 4(R\kappa^{2r} + \sqrt{M})\left(\frac{\kappa}{m\sqrt{\widetilde{\lambda}}} + \frac{\sqrt{2\sqrt{\nu}c_\gamma}}{\sqrt{m\widetilde{\lambda}^\gamma}}\right)\log\frac{8n}{\delta}.$$

Applying a union bound so the following holds with probability at least $1 - \sum_{v\in V}\delta'_v = 1 - \frac{\delta}{2}$ for any $k \in \mathbb{N}$:

$$\max_{w\in V}\left\{\|\mathcal{T}_{\rho,\widetilde{\lambda}}^{-1/2}N_{k,w}\|_H\right\} \leq 4(R\kappa^{2r} + \sqrt{M})\left(\frac{\kappa}{m\sqrt{\widetilde{\lambda}}} + \frac{\sqrt{2\sqrt{\nu}c_\gamma}}{\sqrt{m\widetilde{\lambda}^\gamma}}\right)\log\frac{8n}{\delta}$$

$$\leq \frac{4(R\kappa^{2r} + \sqrt{M})(2\kappa + \sqrt{2\sqrt{\nu}c_\gamma})\log\frac{8n}{\delta}}{\sqrt{(m\widetilde{\lambda})\wedge\widetilde{\lambda}^\gamma}\sqrt{m}}, \tag{17}$$

where we used that $\kappa \geq 1$. The terms $\mathbf{B}_1$ and $\mathbf{B}_2$ are now bounded in the following two paragraphs.

**Term $\mathbf{B}_1$**   Using the high-probability bound (17), the following holds with probability at least $1 - \frac{\delta}{2}$:

$$\mathbf{B}_1 \leq \|\mathcal{T}_\rho^{1/2}\mathcal{T}_{\rho,\lambda}^{1/2}\|\|\mathcal{T}_{\rho,\widetilde{\lambda}}^{1/2}\|\frac{4(R\kappa^{2r} + \sqrt{M})(2\kappa + \sqrt{2\sqrt{\nu}c_\gamma})}{\sqrt{(m\widetilde{\lambda})\wedge\widetilde{\lambda}^\gamma}}\frac{\log\frac{8n}{\delta}}{\sqrt{m}}\eta_t\sum_{k=1}^{t-1}\eta_k$$

$$\leq \|\mathcal{T}_\rho^{1/2}\mathcal{T}_{\rho,\lambda}^{1/2}\|\|\mathcal{T}_{\rho,\widetilde{\lambda}}^{1/2}\|\frac{4(R\kappa^{2r} + \sqrt{M})(2\kappa + \sqrt{2\sqrt{\nu}c_\gamma})}{\sqrt{(m\widetilde{\lambda})\wedge\widetilde{\lambda}^\gamma}(1-\theta)}\frac{\log\frac{8n}{\delta}}{\sqrt{m}}t^{-1}(\eta t^{1-\theta})^2,$$

where we have applied the integral bound t $\sum_{k=1}^{t-1}k^{-\theta} \leq \frac{t^{1-\theta}}{1-\theta}$, see for instance [27, Lemma 12], on the following summation:

$$\eta_t\sum_{k=1}^{t-1}\eta_k = \eta^2 t^{-\theta}\sum_{k=1}^{t-1}k^{-\theta} \leq \frac{\eta^2}{1-\theta}t^{1-2\theta} = \frac{t^{-1}(\eta t^{1-\theta})^2}{1-\theta}.$$

**Term $\mathbf{B}_2$**   Similarly, using the high-probability bound (17), the following holds with probability at least $1 - \frac{\delta}{2}$:

$$\mathbf{B}_2 \leq \|\mathcal{T}_{\rho,\widetilde{\lambda}}^{1/2}\|\frac{4(R\kappa^{2r} + \sqrt{M})(2\kappa + \sqrt{2\sqrt{\nu}c_\gamma})}{\sqrt{(m\widetilde{\lambda})\wedge\widetilde{\lambda}^\gamma}}\frac{\log\frac{8n}{\delta}}{\sqrt{m}}$$

$$\times\sum_{k=1}^{t-2}\eta_k\sum_{j=k+1}^{t-1}\eta_j\left(\frac{1}{\sum_{i=j+1}^t\eta_i} + \left(\frac{\lambda}{\sum_{i=j+1}^t\eta_i}\right)^{1/2}\right).$$

We proceed to bound the remaining terms by utilising results from Section C.5. Firstly, switching the order of sums and applying an integral bound yields

$$\sum_{k=1}^{t-2} \eta_k \sum_{j=k+1}^{t-1} \frac{\eta_j}{\sum_{i=j+1}^{t} \eta_i} = \eta \sum_{k=1}^{t-2} k^{-\theta} \sum_{j=k+1}^{t-1} \frac{j^{-\theta}}{\sum_{i=j+1}^{t} i^{-\theta}}$$

$$= \eta \sum_{j=2}^{t-1} \frac{j^{-\theta}}{\sum_{i=j+1}^{t} i^{-\theta}} \sum_{k=1}^{j-1} k^{-\theta}$$

$$\leq \frac{\eta}{1-\theta} \sum_{j=2}^{t-1} \frac{j^{-\theta}(j-1)^{1-\theta}}{\sum_{i=j+1}^{t} i^{-\theta}}. \tag{18}$$

At this point use $\sum_{i=j+1}^{t} i^{-\theta} \geq t^{-\theta}(t-j)$ as well as Lemma 10 to obtain

$$\sum_{j=2}^{t-1} \frac{j^{-\theta}(j-1)^{1-\theta}}{\sum_{i=j+1}^{t} i^{-\theta}} \leq t^{\theta} \sum_{j=2}^{t-2} \frac{(j-1)^{1-2\theta}}{t-j} \leq 4t^{\theta} t^{-\min(2\theta-1,1)} \log(t) = 4t^{1-\theta} \log(t).$$

For the second term follow the steps to (18) and use Lemma 11 as follows:

$$\sum_{k=1}^{t-2} \eta_k \sum_{j=k+1}^{t-1} \frac{\eta_j}{\left(\sum_{i=j+1}^{t} \eta_i\right)^{1/2}} \leq \frac{\eta^{3/2} t^{\theta/2}}{1-\theta} \sum_{j=2}^{t-1} \frac{(j-1)^{1-2\theta}}{(t-j)^{1/2}}$$

$$\leq \frac{4\eta^{3/2} t^{\theta/2}}{1-\theta} t^{\max(3/2-2\theta,0)}$$

$$= \frac{4\eta^{3/2}}{1-\theta} t^{\max(3(1-\theta)/2,\theta/2)}.$$

This results in the following bound for $\mathbf{B}_2$, which holds with probability at least $1 - \frac{\delta}{2}$:

$$\mathbf{B}_2 \leq \|\mathcal{T}_{\rho,\widetilde{\lambda}}^{1/2}\| \frac{4(R\kappa^{2r} + \sqrt{M})(2\kappa + \sqrt{2\sqrt{\nu}c_\gamma})}{\sqrt{(m\widetilde{\lambda}) \wedge \widetilde{\lambda}^\gamma}(1-\theta)} \frac{\log \frac{8n}{\delta} \log(t)}{\sqrt{m}} \left(4\eta t^{1-\theta} + 4\sqrt{\lambda}\left(\eta t^{\max(1-\theta,\theta/3)}\right)^{3/2}\right).$$

The final bound arises by bringing everything together with a union bound implying it holds with probability at least $1 - \frac{\delta}{2} - \frac{\delta}{2} = 1 - \delta$. Constants are then cleaned up using $\lambda \leq \|\mathcal{T}_\rho\|$ as well as $\widetilde{\lambda} \leq \|\mathcal{T}_\rho\|$ to say $\|\mathcal{T}_\rho^{1/2}\mathcal{T}_{\rho,\widetilde{\lambda}}^{1/2}\|\|\mathcal{T}_{\rho,\widetilde{\lambda}}^{1/2}\| \leq 4\|\mathcal{T}_\rho\|^{3/2}$ and $\|\mathcal{T}_{\rho,\widetilde{\lambda}}^{1/2}\| \leq 2\|\mathcal{T}_\rho\|^{1/2}$. $\qquad\square$

### C.3.3 Network Error bound

In this section we bring together the bounds developed in the previous two sections for the **Population Covariance Error** term and **Residual Empirical Covariance Error** term to construct the final bound on the Network Term as presented in the following theorem.

**Theorem 4.** *Let Assumptions 1, 2, 3 hold with $r \geq 1/2$, and $\eta_t = \eta t^{-\theta}$ for $t \in \mathbb{N}$ with $\eta\kappa^2 \leq 1$ and $\theta \in (0, 3/4)$. Assume $t/2 \geq \lceil \frac{(r+1)\log(t)}{1-\sigma_2} \rceil =: t^\star$ The following bound holds for any $v \in V$, $\alpha \in [0, 1/2]$ and $\gamma' \in [1, \gamma]$:*

$$\mathbf{E}[\|\mathcal{S}_\rho(\omega_{t+1,v} - \xi_{t+1,v})\|_\rho^2] \leq 2\frac{\widetilde{a}\log^2(4n)\log^2(t^\star)}{m}\left(\eta^2 t^{-2r} \vee (m^{-1}(\eta t^\star)^{1+2\alpha}) \vee (\eta t^\star)^{\gamma'+2\alpha}\right)$$

$$+ 2\widetilde{b}_2 \frac{\log^4(8n)\log^2(t)}{m^2}\left(1 \vee (\eta t^{1-\theta})^2 \vee (t^{-2}(\eta t^{1-\theta})^4)\right)\left((m^{-1}\eta t^{1-\theta}) \vee (\eta t^{1-\theta})^\gamma\right),$$

*where $\widetilde{b}_2 = 64\frac{(\|\mathcal{T}_\rho\|+1)^2}{(\|\mathcal{T}_\rho\|\wedge\|\mathcal{T}_\rho\|^\gamma)^2}\widetilde{b}_1^2$ with $\widetilde{b}_1$ defined as in Theorem 9 and $\widetilde{a}$ defined as in Lemma 8.*

*Proof.* Use decomposition (9). Taking the expectation, note that the first term $\mathbf{E}[(\mathbf{Pop.\ Cov.\ Error})^2]$ is controlled by Lemma 8. We now proceed to control the term $\mathbf{E}[(\mathbf{Resid.\ Emp.\ Cov.\ Error})^2]$.

Begin by using the high-probability bound for **Resid. Emp. Cov. Error** in Lemma 9, with $\widetilde{\lambda} = \|\mathcal{T}_\rho\|$ and $\lambda = \|\mathcal{T}_\rho\|(\eta t^{1-\theta})^{-1}$. The following upper bound holds for the quantity that appears in Lemma 9:

$$\frac{1}{(m\lambda) \wedge \lambda^\gamma} = \frac{1}{(\|\mathcal{T}_\rho\| m (\eta t^{1-\theta})^{-1}) \wedge (\|\mathcal{T}_\rho\|^\gamma (\eta t^{1-\theta})^{-\gamma})}$$

$$\leq \frac{1}{\|\mathcal{T}_\rho\| \wedge \|\mathcal{T}_\rho\|^\gamma} \left( (m^{-1}(\eta t^{1-\theta})) \vee (\eta t^{1-\theta})^\gamma \right).$$

Plugging the above into Lemma 9 for the **Resid. Emp. Cov. Error** allows the expectation to be bounded with Lemma 4:

$$\mathbf{E}[(\textbf{Resid. Emp. Cov. Error})^2]$$

$$\leq \widetilde{b}_1^2 \frac{(\|\mathcal{T}_\rho\| + 1)^2}{(\|\mathcal{T}_\rho\| \wedge \|\mathcal{T}_\rho\|^\gamma)^2} \frac{\log^2(t)}{m^2} \left(1 \vee (\eta t^{1-\theta})^2 \vee (t^{-2}(\eta t^{1-\theta})^4)\right)\left((m^{-1}\eta t^{1-\theta}) \vee (\eta t^{1-\theta})^\gamma\right)$$

$$\times \int_0^1 \log^4 \frac{8n}{\delta} d\delta.$$

The result is arrived at by using $\int_0^1 \log^4 \frac{8n}{\delta} d\delta \leq 64 \log^4(8n)$ and bringing together the two bounds for $\mathbf{E}[(\textbf{Pop. Cov. Error})^2]$ and $\mathbf{E}[(\textbf{Resid. Emp. Cov. Error})^2]$. $\qquad\square$

## C.4 Final Bound

In this section we bring together the bounds from the previous sections to construct the final bounds in Theorem 2 and Theorem 1 in the main body of the work. The main result is the following.

**Theorem 5.** *Let Assumptions 1, 2, 3 hold with $r \geq 1/2$ and $\eta_t = \eta t^{-\theta}$ for all $t \in \mathbb{N}$ with $\eta \kappa^2 \leq 1$ $\theta \in (0, 3/4)$. The following holds for all $t/2 \geq \lceil \frac{(r+1)\log(t)}{1-\sigma_2} \rceil =: t^\star$, any $v \in V$, $\alpha \in [0, 1/2]$ and $\gamma' \in [1, \gamma]$:*

$$\mathbf{E}[\mathcal{E}(\omega_{t+1,v})] - \inf_{\omega \in H} \mathcal{E}(\omega) \leq 2R^2(\eta t^{1-\theta})^{-2r}$$

$$+ \widetilde{d}_4 (nm)^{-2r/(2r+\gamma)} \left(1 \vee (nm)^{-2/(2r+\gamma)} (\eta t^{1-\theta})^2 \vee t^{-2}(\eta t^{1-\theta})^2\right) \log^2(t)$$

$$+ 8 \frac{\widetilde{a} \log^2(4n) \log^2(t^\star)}{m} \left( \eta^2 t^{-2r} \vee (m^{-1}(\eta t^\star)^{1+2\alpha}) \vee (\eta t^\star)^{\gamma'+2\alpha} \right)$$

$$+ 8 \frac{\widetilde{b}_2 \log^4(8n) \log^2(t)}{m^2} \left(1 \vee (\eta t^{1-\theta})^2 \vee t^{-2}(\eta t^{1-\theta})^4\right)\left((m^{-1}\eta t^{1-\theta}) \vee (\eta t^{1-\theta})^\gamma\right),$$

*where $\widetilde{d}_4 = 4\left(\frac{2r+\gamma}{2r+\gamma-1}\right)^2 \widetilde{d}_3^2$ with $\widetilde{d}_3$ defined as in Theorem 3.*

*Proof.* Begin with the decomposition in Proposition 1 and take the expectation $\mathbf{E}[\cdot]$. Plug in the bounds for each term proven in the previous sections, i.e. Proposition 2 for the Bias, Theorem 3 with $p = 1/(2r+\gamma)$ for the Sample Variance term and Theorem 4 for the Network Error term. $\qquad\square$

Theorem 2 follows directly from Theorem 5.

*Proof of Theorem 2.* Consider Theorem 5 with constants

$$q_1 = 2R^2$$
$$q_2 = \widetilde{d}_4$$
$$q_3 = 16\widetilde{a}(\log^2(4) + 1)$$
$$q_4 = 24\widetilde{b}_2(\log^2(8) + 1)^2,$$

where the sample variance constant $\widetilde{d}_4$ is defined in Theorem 5, the first network error constant $\widetilde{a}$ is defined in Lemma 8, and the second network error constant $\widetilde{b}_2$ is defined in Theorem 4. $\qquad\square$

We now go on to prove Theorem 1.

*Proof of Theorem 1.* Consider the setting of Theorem 5 with $\theta = 0$. Begin by setting

$$t = \left\lceil (nm)^{1/(2r+\gamma)} \left[ \frac{1}{1-\sigma_2}\left(\frac{n^r}{m^{r+\gamma}}\right)^{2/((1+2\alpha)(2r+\gamma))} \vee \frac{1}{1-\sigma_2}\left(\frac{n^{2r}}{m^\gamma}\right)^{1/((\gamma'+2\alpha)(2r+\gamma))} \vee 1 \right] \right\rceil$$

and $\eta = \kappa^{-2}(nm)^{1/(2r+\gamma)}/t$. It is clear that $\eta t = \kappa^{-2}(nm)^{1/(2r+\gamma)}$. We proceed to show that this choice of iterations $t$ and step size $\eta$ ensures each of the terms in the bound of Theorem 5 are of order $\widetilde{O}((nm)^{-2r/(2r+\gamma)})$.

The Bias term is

$$2R^2(\eta t)^{-2r} = 2R^2\kappa^{4r}(nm)^{-2r/(2r+\gamma)}.$$

The Sample Variance term is bounded as follows:

$$\widetilde{d}_4(nm)^{-2r/(2r+\gamma)}\left(1 \vee (nm)^{-2/(2r+\gamma)}(\eta t)^2 \vee t^{-2}(\eta t)^2\right)\log^2(t)$$

$$\leq 4\kappa^{-4}\widetilde{d}_4(nm)^{-2r/(2r+\gamma)}\log^2(t).$$

The first Network Error term is bounded in three parts aligning with the three terms within the quantity $m^{-1}(\eta^2 t^{-2r} \vee (m^{-1}(\eta t^\star)^{1+2\alpha}) \vee (\eta t^\star)^{\gamma'+2\alpha}))$. Firstly, as $t \geq (nm)^{1/(2r+\gamma)}$ and $\eta \leq 1/\kappa^2$ we get $\eta^2 t^{-2r} \leq \kappa^{-4}(nm)^{-2r/(2r+\gamma)}$. Secondly, from $t \geq (nm)^{1/(2r+\gamma)}\frac{1}{1-\sigma_2}\left(\frac{n^r}{m^{r+\gamma}}\right)^{2/((1+2\alpha)(2r+\gamma))}$ ensuring $\eta \leq \kappa^{-2}(1-\sigma_2)\left(\frac{m^{r+\gamma}}{n^r}\right)^{2/((1+2\alpha)(2r+\gamma))}$ we get

$$\frac{(\eta t^\star)^{1+2\alpha}}{m^2} \leq (\kappa^{-2}2(r+1)\log(t))^{1+2\alpha}\frac{m^{2(r+\gamma)/(2r+\gamma)-2}}{n^{2r/(2r+\gamma)}}$$

$$= (\kappa^{-2}2(r+1)\log(t))^{1+2\alpha}(nm)^{-2r/(2r+\gamma)}.$$

Thirdly, from $t \geq (nm)^{1/(2r+\gamma)}\frac{1}{1-\sigma_2}\left(\frac{n^{2r}}{m^\gamma}\right)^{1/((\gamma'+2\alpha)(2r+\gamma))}$ we have $\eta \leq \kappa^{-2}(1-\sigma_2)\left(\frac{m^\gamma}{n^{2r}}\right)^{1/((\gamma'+2\alpha)(2r+\gamma))}$ and so

$$\frac{(\eta t^\star)^{\gamma'+2\alpha}}{m} \leq (\kappa^{-2}2(r+1)\log(t))^{\gamma'+2\alpha}\frac{m^{\gamma/(2r+\gamma)-1}}{n^{2r/(2r+\gamma)}}$$

$$= (\kappa^{-2}2(r+1)\log(t))^{\gamma'+2\alpha}(nm)^{-2r/(2r+\gamma)}.$$

Using the above three bounds we arrive at the first Network term being $\widetilde{O}((nm)^{-2r/(2r+\gamma)})$.

Now consider the second Network Error term. Since $\eta t = \kappa^{-2}(nm)^{1/(2r+\gamma)}$ and $m \geq n^{\frac{2r+2+\gamma}{2r+\gamma-2}} \geq n^{\frac{1-\gamma}{2(r+\gamma)-1}}$ we have

$$\left(1 \vee (\eta t)^2 \vee t^{-2}(\eta t)^4\right)\left((m^{-1}(\eta t)) \vee (\eta t)^\gamma\right) \leq \left(1 \vee (\eta t)^{2+\gamma} \vee t^{-2}(\eta t)^{4+\gamma}\right).$$

The second Network Error term then becomes, due to $t \geq (nm)^{1/(2r+\gamma)}$,

$$8\frac{\widetilde{b}_2\log^4(8n)\log^2(t)}{m^2}\left(1 \vee (\eta t)^{2+\gamma} \vee t^{-2}(\eta t)^{4+\gamma}\right)$$

$$\leq 8(\kappa^{-2})^{2+\gamma}\widetilde{b}_2\log^4(8n)\log^2(t)\frac{(nm)^{(2+\gamma)/(2r+\gamma)}}{m^2}.$$

For this quantity to be $\widetilde{O}((nm)^{-2r/(2r+\gamma)})$ we require $\frac{(nm)^{(2+\gamma)/(2r+\gamma)}}{m^2} \leq (nm)^{-2r/(2r+\gamma)}$ which is satisfied for $m \geq n^{(2r+\gamma+2)/(2r+\gamma-2)}$. Now ensure $\frac{t}{\log(t)} \geq 2\frac{(1+r)}{1-\sigma_2}$. Note the previous requirements on the iterations $t$ imply

$$t \geq \frac{(nm)^{1/(2r+\gamma)}}{1-\sigma_2}\frac{n^{2r/(2r+\gamma)}}{m^{\gamma/(2r+\gamma)}} \geq \frac{n^{(2r+1)/2r+\gamma}}{1-\sigma_2} \geq \frac{n}{1-\sigma_2}.$$

And since $x \to x/(\log(x))$ is increasing for $x \geq 1$, the requirement $t \geq 2\frac{(1+r)\log(t)}{(1-\sigma_2)}$ is satisfied by $\frac{n}{\log(\frac{n}{1-\sigma_2})} \geq 2(1+r)$.

Now, consider choosing $\gamma' \in [1, \gamma]$ and $\alpha \in [0, 1/2]$ to minimise the number of iterations $t$. Consider the two cases $m \geq n^{2r/\gamma}$ and $m \leq n^{2r/\gamma}$. When $m \geq n^{2r/\gamma}$ we have both $\frac{n^{2r}}{m^\gamma} \leq 1$ and $\frac{n^r}{m^{r+\gamma}} \leq 1$ so the number of iterations $t$ required is minimised by picking $\gamma' = \gamma$ and $\alpha = 0$. Since $2(r + \gamma) \geq 1$ we get $\frac{n^{2r}}{m^{2(r+\gamma)}} \leq \frac{n^{2r/\gamma}}{m}$ and the number of iterations becomes

$t = (nm)^{1/(2r+\gamma)}\left[\left(\frac{1}{1-\sigma_2}\left(\frac{n^{2r/\gamma}}{m}\right)^{1/(2r+\gamma)}\right) \vee 1\right] = (nm)^{1/(2r+\gamma)}\left[\left(\frac{(nm)^{2r/(2r+\gamma)}}{m(1-\sigma_2)^\gamma}\right)^{1/\gamma} \vee 1\right]$. When

$\frac{n^{2r}}{m^\gamma} \geq 1$, the number of iterations $t$ required is minimised by: setting $\gamma' = 1$, noting $\frac{n^{2r}}{m^{2(r+\gamma)}} \leq \frac{n^{2r}}{m^\gamma}$ and further picking $\alpha = 1/2$. It is clear in this case that the number of iterations required becomes

$t = (nm)^{1/2r+\gamma}\frac{1}{1-\sigma_2}\left(\frac{n^r}{m^{\gamma/2}}\right)^{1/(2r+\gamma)} = (nm)^{1/(2r+\gamma)}\frac{(nm)^{r/(2r+\gamma)}}{\sqrt{m}(1-\sigma_2)}$.

$\square$

## C.5 Useful inequalities

In this section we collect useful inequalities used within the proofs.

**Lemma 10.** *The following holds for $q \in \mathbb{R}$ and $t \in \mathbb{N}$ with $t \geq 3$:*

$$\sum_{k-1}^{t-1}\frac{1}{t-k}k^{-q} \leq 2t^{-\min(q,1)}(1 + \log(t)).$$

*Proof.* See Lemma 14 in [27]. $\square$

**Lemma 11.** *The following holds for $q \in \mathbb{R}$ and $t \in \mathbb{N}$ with $t \geq 3$:*

$$\sum_{k-1}^{t-1}\frac{1}{(t-k)^{1/2}}k^{-q} \leq 4t^{\max(1/2-q,0)}.$$

*Proof.* Begin with

$$\sum_{k=1}^{t-1}\frac{1}{(t-k)^{1/2}}k^{-q} \leq t^{\max(1/2-q,0)}\sum_{k=1}^{t-1}\frac{1}{(t-k)^{1/2}k^{1/2}}.$$

Suppose $t$ is even. The bound arises by splitting the sum and using the integral bounds

$$\sum_{k=1}^{t/2}\frac{1}{(t-k)^{1/2}k^{1/2}} \leq \frac{\sqrt{2}}{t^{1/2}}\sum_{k=1}^{t/2}\frac{1}{k^{1/2}} \leq \frac{\sqrt{2}}{t^{1/2}}\left[1 + \int_1^{t/2}x^{-1/2}dx\right] = \frac{\sqrt{2}}{t^{1/2}}\left[1 + 2\left(\sqrt{\frac{t}{2}} - 1\right)\right] \leq 2,$$

and

$$\sum_{k=t/2+1}^{t-1}\frac{1}{(t-k)^{1/2}k^{1/2}} \leq \sqrt{\frac{2}{t}}\sum_{k=t/2+1}^{t-1}\frac{1}{(t-k)^{1/2}} \leq \sqrt{\frac{2}{t}}\left[1 + \int_{t/2+1}^{t-1}(t-x)^{-1/2}dx\right]$$

$$= \sqrt{\frac{2}{t}}\left[1 + 2\left(\sqrt{\frac{t}{2}-1} - 1\right)\right] \leq 2.$$

If $t$ is odd, follow the steps above and split the sum at $k = (t-1)/2$ and $k = (t-1)/2 + 1$. $\square$

## Footnotes

[3] The units of time within [50, Section 3.2] are in terms of the time taken to compute a gradient for $nm$ samples, and as such, can be translated into units per gradient computation by multiplying by $nm$.

[4] The operator norm can be bounded $\|\mathcal{T}_\rho^{1/2} \Pi_{t:k+1}(\mathcal{T}_\rho) \mathcal{T}_{\rho,\lambda}^{1/2}\| \leq \sup_{x \in (0,\kappa^2)} \{x^{1/2}(x+\lambda)^{1/2} \prod_{\ell=k+1}^t (1 - \eta_\ell x)\} \leq \sup_{x \in (0,\kappa^2)} \{x \prod_{\ell=k+1}^t (1 - \eta_\ell x)\} + \sqrt{\lambda} \sup_{x \in (0,\kappa^2)} \{x^{1/2} \prod_{\ell=k+1}^t (1 - \eta_\ell x)\}$. Using techniques used to prove [27, Lemma 15], these terms can be bounded as shown.

[5] For an operator $L$ note that $\|L\| = \|LL^\star\|^{1/2}$ where $L^\star$ is the adjoint of $L$. The Hilbert-Schmidt norm bounds the operator norm as we have $\|L\|^2 = \|LL^\star\| \leq \mathrm{Tr}\left(LL^\star\right) = \|L\|_{HS}^2$.