[Reviews · NeurIPS 2019]

Reviewer 1



[I have read the author response; no change in score as my confidence is the limiting factor] In eq (3), it seems to me that "m" was not introduced - it is only in words defined later. As I was reading the result, I was thinking about this question: "What is the number of samples needed for optimal rates, if the communication network is fully connected graph?" This is an interesting special case, which should be implied by existing work, probably worth highlighting in the paper. How does that special case compare to the presented result? I imagine there is a gap. If so, this would open another point for future directions - presented are two extremes - but how does the threshold depend on network topology?

Reviewer 2



This paper proves optimal generalization error bounds for Distributed Gradient Descent in the context of multi-agent decentralized non-parametric least-squares regression when i.i.d. samples are assigned to agents. The derived results show that if the agents hold sufficiently many samples with respect to the network size, then Distributed Gradient Descent with a number of iterations (depending on a network parameter) achieves optimal rates. Provided the “communication delay” is sufficiently small, the Distributed Gradient Descent yields a linear speed-up in runtime compared to the single machine protocol. To the best of my knowledge, the derived result is the first statistical result for Distributed Gradient Descent in the context of multi-agent decentralized non-parametric least-squares regression, making an important contribution for understanding the generalization properties of decentralized distributed gradient descent. Overall, I find the results are very interesting. The novel key step of the proof is the estimation of the network error. I did a high-level check of this proof step and I did not find anything wrong. The presentation is fine, although I believe it can be further be improved (for example, making more concise in some of the statements). Minor comments: Line 145 of Page 4, the marginal distribution on X… The presentation in the supplementary material could be further improved. For example, the notation $T_{\rho}$ is not introduced in Line 93 on Page 3. In the appendix, the proof involves many notations. But it seems to me these notations are necessary. Could you derive similar results for the non-attainable cases? --------After Rebuttal---------------- I have read the rebuttal. I am happy with the acceptance of the paper.

Reviewer 3



Update: I have read the author response and appreciate that they addressed some of my comments. I'm leaving my score unchanged, mainly due to the remaining limitations. -- This paper studies the performance of decentralized gradient descent for nonparametric (linear) regression (in a RKHS). The focus is on obtaining statistical guarantees about the generalization. This is a highly relevant direction to the growing body of work on decentralized training. The paper is generally well written, contains very original ideas, and I was very excited to read it. The main reason I didn't give a higher rating was because of the limitations listed at the beginning of Sec 5. I commend the authors for acknowledging them. Nevertheless, they make the reader question the relevance of the results (e.g., not covering the case of finite-dimensional linear regression). I only have a few suggestions/questions. 1. In line 254 it may be worth emphasizing that the relaxation time 1/(1 - \sigma_2) in the setting considered (static, symmetric doubly-stochastic P) is at worst polynomial in n, so the condition on n is no very restrictive. 2. In line 275, please elaborate on the communication model. I would have expected the denominator on the right to involve the product of \tau and Deg(P), or otherwise for \tau to at least be increasing in Deg(P), since having to communicate with more neighbors should require more communication. How would that affect the conclusions in Sec 3.1? Aside: Such a model would be similar to the one considered in Tsianos and Rabbat, NIPS 2012, which also investigated protocols which progressively sparsify communication over time (relevant to the discussion in Sec 5). 3. The limitations list at the beginning of Sec 5 seem pretty severe. Could you elaborate why modified bounds for the finite-dimensional setting would only hold for "easy" problems, and what is the challenge in generalizing these results to other finite-dimensional problems? (Note: It seems questionable to focus on the infinite dimensional setting for decentralized algorithms, since it is no longer clear how to communicate infinite-dimensional quantities using a finite number of bits, so that the communication delay is finite.) 4. The result in Sec 3.1 begins to hint at a tradeoff in terms of the parameter \tau. I would have liked to see more discussion around this. In particular, for very large \tau, should a regime appear where it is most efficient for all nodes to work independently (for m sufficiently large)? 5. Regarding accelerated gossip, results similar to those mentioned are also proved in Oreshkin, Coates, and Rabbat, "Optimization and analysis of distributed averaging with short node memory," IEEE Trans Signal Processing, 2010.

[Author Response · NeurIPS 2019]

We thanks the reviewers for their feedback. We now proceed to reply to their comments.

**Number of Samples for Complete Graph? (R1)**: In this case there is no network error and all of the agents iterates
are identical to centralised single-machine Gradient Descent (GD) applied to all of the samples within the network,
as we point out in Appendix B.1. Error Decomposition. There is no condition on the number of samples to achieve
optimal statistical rates in this case [25].

**Extending Current Limitations (R2, R3)**:
In light of the comments we have pursed extending the theory on the following fronts:

- **Non-attainable case**: Consider the capacity independent ($\gamma = 1$) non-attainable case ($r < 1/2$), when
agents know the population covarince $\mathcal{T}_\rho$ i.e. additive noise oracle or fixed-design regression [15,20]. The
network error in this case only consists of the population covariance error and can be bounded as follows.
Due to being in the non-attainable case, the bound on the terms $\{N_{k,w_k}\}_{k,w_k}$ now depend on the time
step $k$, in particular with regularisation becoming $\widetilde{O}\big(\frac{1}{\sqrt{m\lambda}} + \frac{(\eta k)^{1/2-r}}{m\sqrt{\lambda}}\big)$, see Lemma 18 in [26]. The first
term matches the attainable case while the second is new and now must be controlled. Following the
analysis for the attainable case, this new term squared can be shown to be $\widetilde{O}\big((\eta t)^{1-2r}(\eta t^\star)/m^2\big)$ in high
probability. With $\eta = O(1)$ and $t = (nm)^{1/(2r+\gamma)}$ this quantity is smaller than the optimal statistical rate
once $m \geq n^{1/(4r+1)}(t^\star)^{(2r+1)/(4r+1)}$. This condition is strictly weaker than $m \geq n^{2r}(t^\star)^{2r+1}$, the condition
that arises in the attainable case to achieve a linear speed up by ensuring the first term is sufficiently small, if
$r \geq 1/4$. We leave the capacity dependent non-attainable case to future work.

- **Residual Covariance Error**: When agents do not know the population covarince, we require $2r + \gamma \geq 2$ to
control the residual covariance error. A consequence in the finite low-dimensional setting ($d < nm, \gamma = 0$)
is that bounds only hold for "easy" problems $r > 1$. We note a similar consequence occurs in the the
high-dimensional setting as source and capacity assumptions analogous to our setting can be utilised to achieve
dimension free bounds (when $d > nm$ the classical $O(d/nm)$ rate is vacuous), see [34]. The limitation
$2r + \gamma \geq 2$ can be improved though sharper control of the residual network error terms, in particular by
repeatedly applying the recursion (Proposition 5) and centering the difference $\Pi_{t:k+1}(\mathcal{T}_{\mathbf{x}_{w_{t:k+1}}}) - \Pi_{t:k}(\mathcal{T}_\rho)$
around the expectation $\mathbf{E}[\Pi_{t:k+1}(\mathcal{T}_{\mathbf{x}_{w_{t:k+1}}})]$.

**Communication Model (R3)**:

- **Description**: We consider a lockstep communication model where each round lasts for $\tau$ units of time. Within
each round, agents send/receive the messages to/from their neighbours in order to implement a single update
of algorithm (3). With a gradient evaluation costing 1 unit of time, each iteration of Distributed Gradient
Descent takes the following amount of time $m + \tau + \text{Deg}(P)$: $m$ gradient evaluations; $\tau$ in communication
delay ; $\text{Deg}(P)$ for each agent to aggregating their neighbours and own gradients, as the sum in algorithm (3)
$\sum_{w \in V} P_{vw}$ has computational cost $O(\text{Deg}(P))$.

- **Communication Delay $\tau$ Increasing with $\text{Deg}(P)$?**: Indeed $\tau$ maybe increasing with the network degree,
although it could depend on other factors arising from: noisy transmission, compressing or decompressing
messages and synchronizing with neighbours. The work Tsianos and Rabbat, NIPS 2012 makes the assumption
that the delay $\tau$ is a linear function of the network degree and transmit time $r \geq 0$ so $\tau = r\text{Deg}(P)$. The
conclusion of Section 3.1 would be unaffected provided $\tau + \text{Deg}(P) = O(m)$.

- **Tradeoff In Terms of the Parameter $\tau$**: For sufficiently large $m$ the speed up is $nm/(m + \tau + \text{Deg}(P))$.
No speed up is achieved if the communication delay is lower bounded $\tau = \Omega(nm)$, that is if it is longer
than the time it takes for centralised single-machine GD to compute a gradient with all of the samples within
the network. More precisely, suppose a network topology with $\text{Deg}(P) = \Theta(n^\beta)$ for $0 \leq \beta \leq 1$, delay
$\tau = r\text{Deg}(P)$ as described previously and $m > n$. Then a linear speed up is achieved if $r \leq m/n^\beta$, no speed
up if $r \geq mn^{1-\beta}$, and a non-linear speed up if $m/n^\beta < r < mn^{1-\beta}$. In contrast, for high degree graphs with
$\beta = 1$ and $(1 - \sigma_2)^{-1} = O(1)$, Sianos and Rabbat, 2012 saw a linear speed up when $r < 1$ whilst we require
$r \leq m/n$. We thank the reviewer for this question as the above is now included within the manuscript.

**Communicating Infinite-Dimensional Quantities (R3)**: A theoretical setting allows precise understanding of the
generalisation capabilties through optimal statistical rates [12]. While particular implementations are outside the scope
of this work, note it is common to use finite approximations of infinite dimensional quantities whilst accounting for the
statistical precision (Learning with sgd and random features Luigi, Rudi & Roasasco 2018).

**Regime Where Nodes Work Independently (R3)** : The objective is for each agent to achieve the optimal statistical
rate with respect to *all* $nm$ data points within the network, and as such, it is a requirement for agents to communicate
one another.

[Meta-Review · NeurIPS 2019]

This paper provides a nice and clean characterization of a decentralized learning problem. The result is perhaps unsurprising in its form, but the analysis is far from trivial. There are some nontrivial assumptions for their results to hold which perhaps limit the scope of this result but do suggest interesting avenues for future research in this increasingly important area. Overall, this is a solid contribution and should be of interest to NeurIPS attendees who work in optimization and distributed systems.